**Greenhouse gases emissions from riparian wetlands: An example from the**
**Inner Mongolia grassland region in China**
**Xinyu Liu[1,2], Xixi Lu[1,3], Ruihong Yu[1,2], Heyang Sun[1], Hao Xue[1], Zhen Qi[1],**
**Zhengxu Cao[1], Zhuangzhuang Zhang[1], Tingxi Liu[4]**
[1] Inner Mongolia Key Laboratory of River and Lake Ecology, School of Ecology and Environment,
Inner Mongolia University, Hohhot 010021, China;
[2] Key Laboratory of Mongolian Plateau Ecology and Resource Utilization, Ministry of Education,
Hohhot 010021, China;
[3] Department of Geography, National University of Singapore, 117570, Singapor;
[4] Inner Mongolia Water Resource Protection and Utilization Key Laboratory, Water Conservancy
and Civil Engineering College, Inner Mongolia Agricultural University, Hohhot 010021, China
**Corresponding author:** Ruihong Yu (rhyu@imu.edu.cn) and Tingxi Liu (txliu@imau.edu.cn)
**Abstract:** Gradual riparian wetland drying is increasingly sensitive to global warming and
contributes to climate change. Riparian wetlands play a significant role in regulating carbon and
nitrogen cycles. In this study, we analyzed the emissions of carbon dioxide ($CO_2$), methane ($CH_4$),
and nitrous oxide ($N_2O$) from riparian wetlands in the Xilin River Basin to understand the role of
these ecosystems in greenhouse gas (GHG) emissions. Moreover, the impact of the catchment
hydrology and soil property variations on GHG emissions over time and space were evaluated.
Our results demonstrate that riparian wetlands emit larger amounts of $CO_2$ (335–2790 mg·m$^{-2}$·h$^{-1}$
in wet season and 72–387 mg·m$^{-2}$·h$^{-1}$ in dry season) than $CH_4$ and $N_2O$ to the atmosphere due to
high plant and soil respiration. The results also reveal clear seasonal variations and spatial patterns
along the transects and in the longitudinal direction. $N_2O$ emissions showed a spatiotemporal
pattern similar to that of $CO_2$ emissions. Near-stream sites were the only sources of $CH_4$
emissions, while the other sites served as sinks for these emissions. Soil moisture content and soil
temperature were the essential factors controlling the GHG emissions, and abundant aboveground
biomass promoted the $CO_2$, $CH_4$, and $N_2O$ emissions. Moreover, compared to different types of
grasslands, riparian wetlands were the potential hotspots of GHG emissions in the Inner
Mongolian region. Degradation of downstream wetlands has resulted in reducing the soil carbon
pool by approximately 60%, reducing $CO_2$ emissions by approximately 35%, and converting the
wetland from a $CH_4$ and $N_2O$ source to a sink. Our study showed that anthropogenic activities
have extensively changed the hydrological characteristics of the riparian wetlands and might
accelerate carbon loss, which could further affect the GHG emissions.

**Key words:** Riparian wetlands, Grasslands, Greenhouse gas, Spatial-temporal distribution, Impact
factor, Xilin River Basin



## 1. Introduction
With the increasing impacts of global warming, the change in the concentrations of
greenhouse gases (GHGs) in the atmosphere is a source of concern in the scientific community
(Cao et al., 2005). According to the World Meteorological Organization (WMO, 2018), the
concentrations of carbon dioxide ($CO_2$), methane ($CH_4$), and nitrous oxide ($N_2O$) have increased
by 146%, 257%, and 122%, respectively, since 1750. Despite their lower atmospheric
concentrations, $CH_4$ and $N_2O$ absorb infrared radiation approximately 28 and 265 times more
effectively at centennial timescales than $CO_2$ (IPCC, 2013). On a global scale, $CO_2$, $CH_4$, and $N_2O$
contribute 87% to the GHG effect (Ferrón et al., 2007).
Wetlands are unique ecosystems that serve as transition zones between terrestrial and aquatic
ecosystems. They play an important role in the global carbon cycle (Beger et al., 2010; Naiman
and Decamps, 1997). Wetlands are sensitive to hydrological changes, particularly in the context of
global climate change (Cheng and Huang, 2016). Moreover, wetland hydrology is affected by
local anthropogenic activities, such as the construction of reservoirs, resulting in gradual drying.
Although wetlands cover only 4–6% of the terrestrial land surface, they contain approximately
12–24% of global terrestrial soil organic carbon (SOC), thus acting as carbon sinks. Moreover,
they release $CO_2$, $CH_4$, and $N_2O$ into the atmosphere and serve as carbon sources (Lv et al., 2013).
In general, the carbon accumulation by plant's photosynthesis is higher than the consumption
(plant respiration, animal respiration, and microbial decomposition) in the wetland, thus the net
effect of the wetland is acted as a carbon sink. Wetlands are increasingly recognized as an
essential part of nature, given their simultaneous functions as carbon sources and sinks. Excessive
rainfall will cause an expansion in wetland areas and a sharp increase in the soil moisture content,
thus enhancing respiration, methanogenesis, nitrification, and denitrification rates (Mitsch et al.,
2009). On the contrary, reduced precipitation or severe droughts will result in a decrease in water
levels, causing the wetlands to dry up. The accumulated carbon will be released back into the
atmosphere through oxidation. Due to the increasing impact of climate change and human activity,
the drying of wetlands has been widely observed in recent years (Liu et al., 2006); more than half
of global wetlands have disappeared since 1900 (Mitsch and Gosselink, 2007), and this tendency
is expected to continue in the future. The loss of wetlands may directly shift the soil environment
from anoxic to oxic conditions, and modify the $CO_2$ and $CH_4$ source and sink functions of wetland
ecological systems (Waddington and Roulet, 2000; Zona et al., 2013).

The Xilin River Basin in China is characterized by a marked spatial gradient in soil moisture

content. It is a unique natural laboratory that may be used to explore the close relationships
between the spatiotemporal variations in hydrology and riparian biogeochemistry. Wetlands
around the Xilin River play an irreplaceable role with regard to local climate control, water
conservation, the carbon and nitrogen cycles, and husbandry (Gou et al., 2015; Kou, 2018).
Moreover, the Xilin River region is subjected to seasonal alterations in precipitation and
temperature regimes, and construction of the Xilin River Reservoir has resulted in highly negative
consequences, such as the drying of downstream wetlands, affecting riparian hydrology as well as
microbial activity in riparian soils. GHG emissions in riparian wetlands vary immensely.
Understanding the interactions between GHG emissions and hydrological changes in the Xilin
River riparian wetlands has thus become increasingly important. Moreover, it is necessary to
estimate the changes in GHG emissions as a result of wetland degradation at local and global
scales.

In this work, GHG emissions from riparian wetlands and adjacent hillslope grasslands of the

Xilin River Basin were investigated. GHG emissions, soil temperature, and soil moisture content
were measured in dry and wet seasons. The main objectives of this study were to (1) investigate
the temporal and spatial variations in $CO_2$, $CH_4$, and $N_2O$ emissions from the wetlands in the
riparian zone, and examine the main factors affecting the GHG emissions, (2) compare the GHG
emissions from the riparian wetlands and different types of grasslands, and (3) evaluate the impact
of wetland degradation in the study area on GHG emissions.

## 2. Materials and methods

### 2.1 Study site



The Xilin River is situated in the southeastern part of the Inner Mongolia Autonomous
Region in China (E115°00'–117°30', N43°26'–44°39'). It is a typical inland river of the Inner
Mongolia grasslands. The river basin area is 10,542 km$^2$, the total length is 268.1 km, and the
average altitude is 988.5 m. According to the meteorological data provided by the Xilinhot
Meteorological Station (Xi et al., 2017; Tong et al., 2004), the long-term annual mean air
temperature is 1.7°C, and the maximum and minimum monthly means are 20.8°C in July and
−19.8°C in January, respectively. The average annual precipitation was 278.9 mm for the period of
1968–2015. Precipitation is distributed unevenly among the seasons, with 87.41% occurring
between May and September.
Soil types in the Xilin River Basin are predominantly chernozems (86.4%), showing a
significant zonal distribution as light chestnut soil, dark chestnut soil, and chernozems from the
northwest to southeast. Soil types in this basin also present a vertical distribution with elevation.
The chernozems are primarily soluble chernozems and carbonate chernozems, distributed at
altitudes above 1350 m with a relatively fertile and deep soil layer. Dark chestnut soil, boggy soil,
and dark meadow with high humus content are distributed between the altitudes of 1150 and 1350
m. Light chestnut soil, saline meadow soil, and meadow solonchak with low soil humus, a thin
soil layer, and coarse soil texture are distributed between the altitudes of 902 and 1150 m (Xi et al.,

2017).

### 2.2 Field measurements and laboratory analyses


In this study, five representative transects were selected as the primary measurement sites in
the entire Xilin River. Each transect cuts through the riparian wetlands near the river and hillslope
grasslands further away from it (Fig. 1).

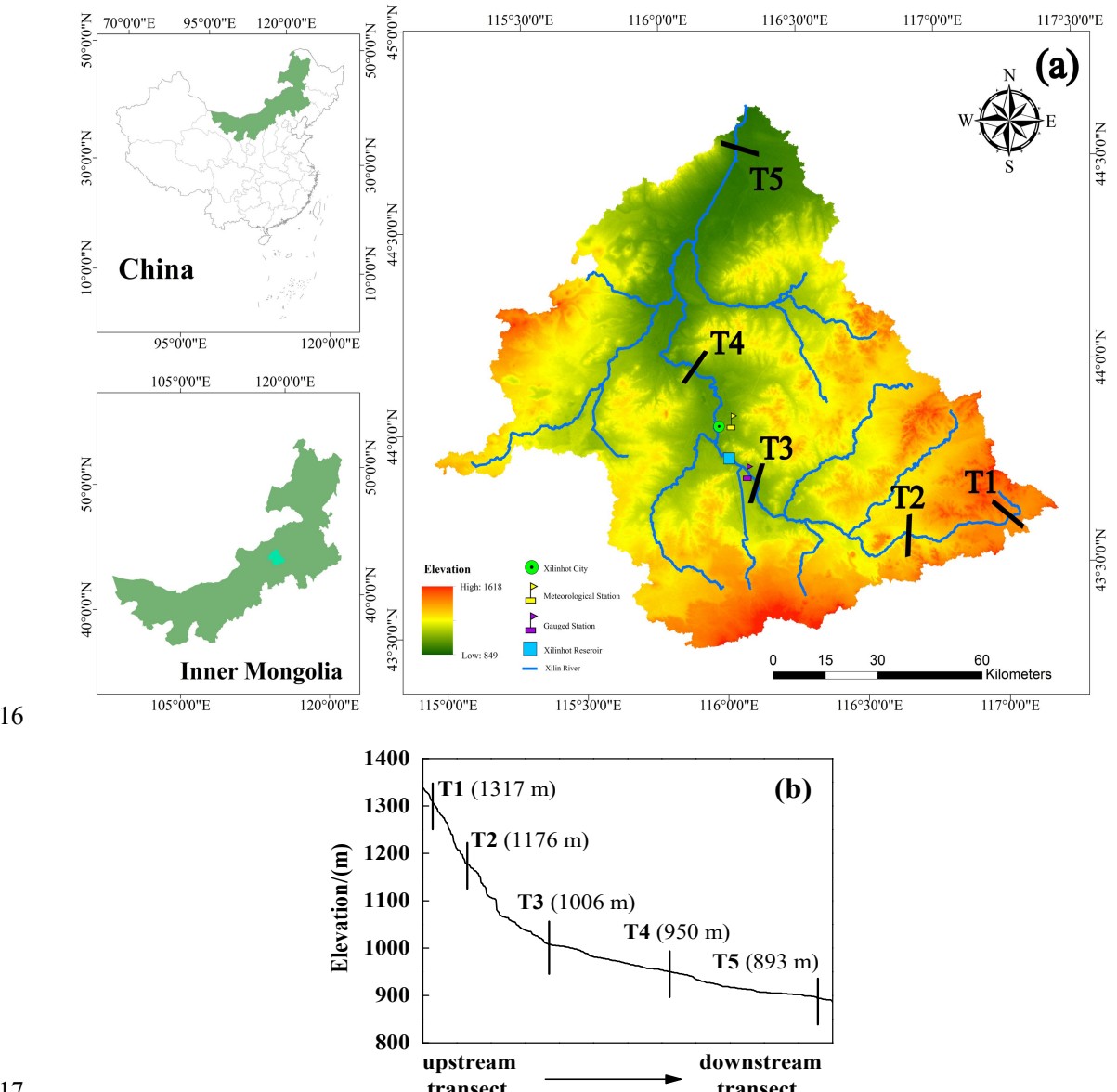



Fig. 1 (a) Location of the Xilin River Basin and distribution of five riparian-hillslope transects

(T1–T5). (b) Elevation details of each transect in the Xilin River Basin.

The layout of the sampling points of each transect is shown in Fig. 2. Each sampling point

from T1–T5 was extended from the river to both sides, to the grassland on the slopes, using 5–7

sampling points for each transect and resulting in 24 points in total. The sampling sites on the left

and right banks were defined as L1–L3 and R1–R4 from the riparian wetlands to the hillslope

grasslands. As transect T3 was located on a much wider flood plain, none of its sampling points

were located on the hillslope grassland. The last transect (T5) was located downstream in the dry

lake and contained seven sampling points. They were defined as S1–S7, where S1, S2, and S7

were located along the lake shore (the lakeside zone), and S3–S6 were located in the dry lake bed
(S3 and S4 in the mudbank, S5 in saline–alkali soil, and S6 in sand–gravel geology). Moreover,
characterizations for T1, T2, and T3 transects were the continuous river flow and T4 and T5
transects were the intermittent river flow.
The $CO_2$, $CH_4$, and $N_2O$ emissions from each site were measured in August (wet season) and
October (dry season) in 2018 using a static dark chamber and the gas chromatography method.
The static chambers were made of a cube-shaped polyvinyl chloride (PVC) pipe (dimensions: 0.4
m × 0.2 m × 0.2 m). A battery-driven fan was installed horizontally inside the top wall of the
chamber to ensure proper air mixing during measurements. To minimize heating from solar
radiation, white adiabatic aluminum foil was used to cover the entire aboveground portion of the
chamber. During measurements, the chambers were driven into the soil to ensure airtightness and
connected with a differential gas analyzer (Li-7000 $CO_2$/$H_2O$ analyzer, LI-COR, USA) to measure
the changes in the soil $CO_2$ concentration. The air in the chamber was sampled using a 60 mL
syringe at 0, 7, 14, 21, and 28 min. The gas samples were stored in a reservoir bag and taken to the
laboratory for $CH_4$ and $N_2O$ measurements using gas chromatography (GC-2030, Japan). The
measurements were scheduled for 9:00–11:00 a.m. or 3:00–5:00 p.m.
Soil temperature (ST) was measured at depths of 0–10 cm and 10–20 cm with a
geothermometer (DTM-461, Hengshui, China). Plant samples were collected in a static chamber
and oven-dried in the laboratory to obtain aboveground biomass (BIO). A 100 $cm^3$ ring cutter was
used to collect surface soil samples at each site, which were placed in aluminum boxes and
immediately brought back to the laboratory to measure soil mass moisture content (SMC) and soil
bulk density ($\rho_b$) using national standard methods (NATESC, 2006). Topsoil samples were
collected, sealed in plastic bags, and brought back to the laboratory to measure soil pH, electrical
conductivity (EC), total soil organic carbon (TOC), and soil C:N ratio.

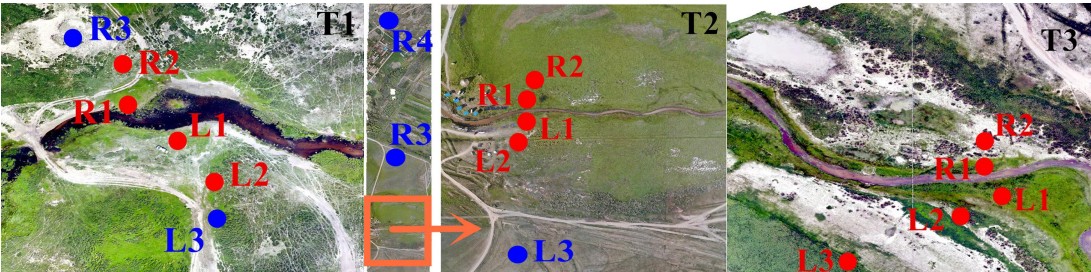


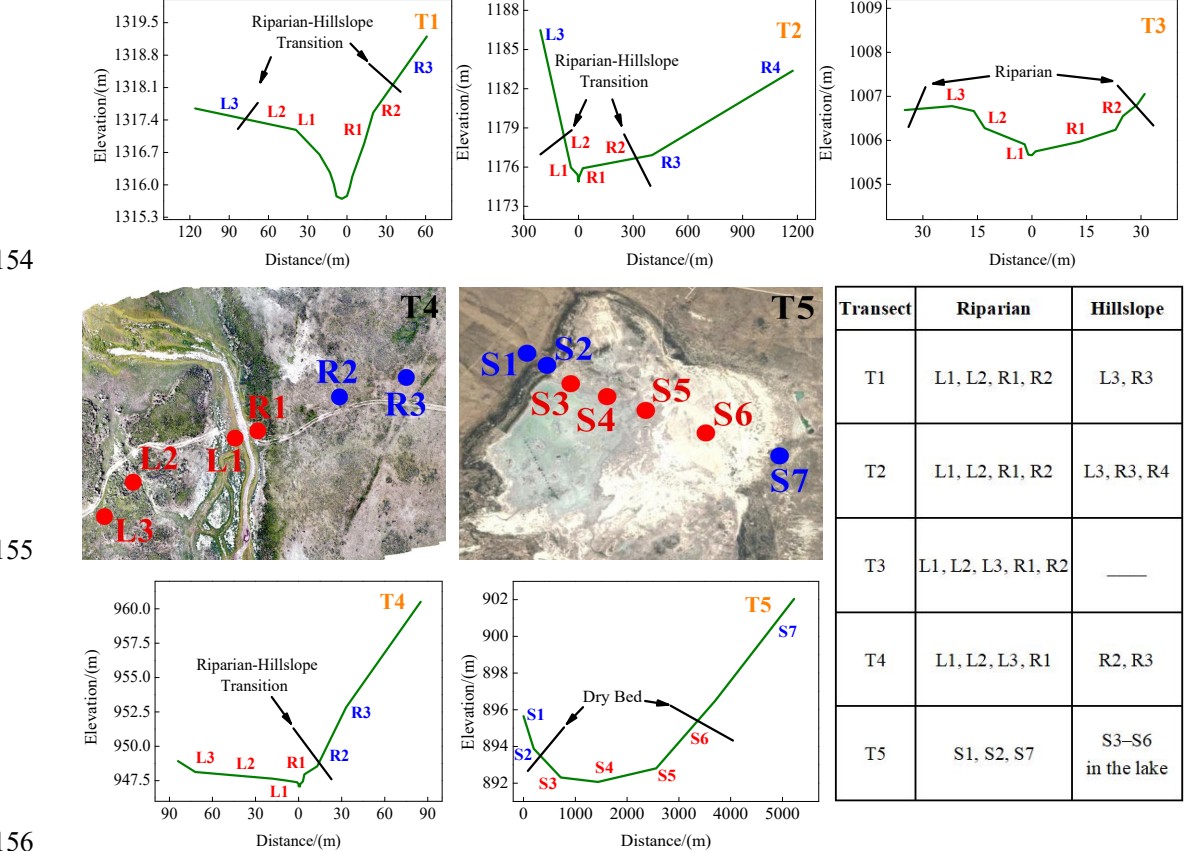





Fig. 2 Distributions of sampling points in transects T1–T5 (The images are authors' own)

Table 1. Physical and chemical properties (Mean±SD) of soils at various sites within each

transect

| Transect | Zone | Samples number | SMC10-V | SMC20-V | Soil C:N | TOC (g·kg⁻¹) | BIO (g) | ρ_b | pH | EC (μs/cm) | SSM (%) |
|---|---|---|---|---|---|---|---|---|---|---|---|
| T1 | Riparian | 12 | 12.16 ± 7.55 | 12.88 ± 12.05 | 12.46 ± 0.91 | 30.16 ± 6.54 | 14.67 ± 5.44 | 1.28 ± 0.07 | 7.25 ± 0.62 | 154.71 ± 23.70 | 47.77 ± 7.04 |
| | Hillslope | 6 | 2.72 ± 0.91 | 5.05 ± 3.09 | 11.41 ± 0.09 | 10.77 ± 4.72 | 6.70 ± 1.48 | 1.45 ± 0.03 | 7.22 ± 0.40 | 82.02 ± 16.37 | 31.02 ± 1.32 |
| T2 | Riparian | 12 | 26.75 ± 19.52 | 12.19 ± 7.82 | 11.70 ± 1.14 | 19.96 ± 5.71 | 24.76 ± 9.65 | 1.23 ± 0.05 | 8.95 ± 0.45 | 303.88 ± 102.16 | 51.21 ± 6.49 |
| | Hillslope | 9 | 5.85 ± 4.82 | 3.03 ± 1.43 | 9.77 ± 0.88 | 14.87 ± 11.21 | 6.10 ± 3.19 | 1.38 ± 0.13 | 8.10 ± 0.55 | 162.97 ± 128.18 | 35.09 ± 6.75 |
| T3 | Riparian | 12 | 28.04 ± 22.95 | 14.53 ± 8.98 | 15.80 ± 4.16 | 22.40 ± 9.69 | 6.37 ± 2.95 | 1.35± 0.19 | 9.50 ± 0.67 | 1233.20 ± 829.83 | 47.56 ± 11.65 |
| | L3 | 3 | 116.37 ± | 113.36 ± | 16.8 ± | 36.1 ± | 107.75 | 0.592 ± | 8.5 ± | 403 ± 57.21 | >100 |

| Transect | Zone | n | | | | | | | | | | |
|---|---|---|---|---|---|---|---|---|---|---|---|---|
| | | | 56.91 | 23.17 | 0.58 | 1.84 | ±16.94 | 0.02 | 0.17 | | |
| T4 | Riparian | 12 | 5.42 ± 3.34 | 4.07 ± 4.31 | 12.52 ± 2.06 | 9.96 ± 1.25 | 11.97 ± 4.50 | 1.30 ± 0.08 | 8.84 ± 0.22 | 461.72 ± 314.27 | 44.08 ± 7.07 |
| | Hillslope | 6 | 3.35 ± 2.06 | 4.27 ± 1.94 | 9.97 ± 0.50 | 9.65 ± 1.05 | 7.84 ± 2.48 | 1.30 ± 0.09 | 8.23 ± 0.14 | 118.5 ± 8.25 | 39.43 ± 5.55 |
| T5 | Dry lake bed | 12 | 17.47 ± 15.08 | 14.49 ± 13.28 | 63.74 ± 12.93 | 31.41 ± 6.55 | 5.48 ± 2.35 | 1.16 ± 0.10 | 9.88 ± 0.18 | 7320.87 ± 4300.03 | 58.47 ± 7.16 |
| | Lake shore | 9 | 2.64 ± 1.48 | 2.82 ± 1.27 | 15.92 ± 4.71 | 6.35 ± 1.16 | 0 | 1.33 ± 0.09 | 9.41 ± 0.7 | 281.82 ± 162.73 | 37.52 ± 5.34 |

Note: SMC10-V - soil volumetric moisture content in 0-10 cm; SMC20-V - soil volumetric moisture content in 10-20 cm; Soil C:N - soil carbon-nitrogen ratio; TOC - total soil organic carbon; BIO - aboveground biomass; $\rho_b$ - soil bulk density; pH - soil pH; EC - soil electrical conductivity; SSM - saturated soil moisture.

Table 2. soil particle composition of soils at various sites within each transect

| Transect | Zone | soil particle composition | | |
|---|---|---|---|---|
| | | Clay % | Silt % | Sand |
| | | (<0.002 mm) | (0.02~0.002 mm) | (2.0 ~0.02 mm) |
| T1 | Riparian | 2.5 | 2.7 | 94.8 |
| | Hillslope | 9.6 | 6.1 | 85.3 |
| T2 | Riparian | 5.5 | 5.8 | 90.7 |
| | Hillslope | 10.8 | 8.6 | 80.6 |
| T3 | Riparian | 4.1 | 1.1 | 94.8 |
| T4 | Riparian | 11.4 | 1.5 | 87.1 |
| | Hillslope | 12.7 | 5.9 | 81.4 |
| T5 | Lake shore | 5.1 | 2.1 | 92.8 |
| | Dry lake bed | 46.1 | 4.8 | 49.1 |

## 2.3 Calculation of GHG emissions

The $CO_2$, $CH_4$, and $N_2O$ emissions were calculated using Eq. 1 (Qin et al., 2016):

$$F = \frac{V}{A} \times \frac{dc}{dt} \times \rho = H \times \frac{dc}{dt} \times \frac{M}{V} \times (\frac{273.15}{273.15+t})$$

(1)

Where $F$ denotes the $CO_2$, $CH_4$, and $N_2O$ emissions (mg·m$^{-2}$·h$^{-1}$), $H$ is the height of the static chamber (0.18 m), $M$ is the relative molecular weight (44 for $CO_2$ and $N_2O$, and 16 for $CH_4$), $V$ is the volume of gas in the standard state (22.4 L·mol$^{-1}$), d$c$/d$t$ is the rate of change of the gas concentration (10$^{-6}$·h$^{-1}$), and $T$ is the temperature in the black chamber (°C).

The annual cumulative emissions were calculated using Eq. 2 (Whiting G and Chanton J.,

2001)

$$M = \sum \frac{F_{i+1} + F_I}{2} \times (t_{i+1} - t_i) \times 24$$
(2)

Where M denotes the total cumulative emissions of $CO_2$, $CH_4$, or $N_2O$ (kg·hm$^2$), $F$ is the emission
flux of $CO_2$, $CH_4$, or $N_2O$, i is the sampling frequency, $t_{i+1}$-$t_i$ represents the interval between two
adjacent measurement dates.
In this study, a 100-year scale was selected to calculate the global warming potential (GWP)
of soil $CH_4$ and $N_2O$ emissions (Whiting G and Chanton J., 2001):
$$GWP = 1 \times [CO_2] + 25 \times [CH_4] + 298 \times [N_2O]$$
(3)

Where 25 and 298 are GWP multiples of $CH_4$ and $N_2O$ relative to $CO_2$ on a 100-year time scale,
respectively.
**2.4 Statistical Analysis**
All statistical analyses were performed using SPSS for Windows version 18.0 (SPSS Inc.,
Chicago, IL, USA). Statistical significance was set at $P < 0.05$. Pearson correlation analysis was
conducted to estimate the relationships between GHGs fluxes and environmental variables. A
Wilcoxon test was used to determine the difference of GHGs fluxes in two seasons.
**3.  Results**
**3.1  Spatiotemporal patterns of SMC for each transect**
The temporal and spatial variations in SMC10 in the following order: wet season > dry
season and riparian wetlands > hillslope grasslands (Fig. 3a, c, e). Similar variations were
observed in SMC20 (Fig. 3b, d, f). The average SMC10 and SMC20 in the continuous river
transects in the riparian zones (37.44% in wet season and 19.40% in dry season; 25.96% in wet
season and 17.39% in dry season) were higher than those in the hillslope grasslands (9.12% in wet
season and 4.15% in dry season; 6.51% in wet season and 5.96% in dry season). During the study
period, both SMC10 and SMC20 changed as the distance from the river increased, and the highest
value was observed at the near-stream sites (L1 and R1). SMC10 fluctuations were low in the
intermittent transect compared to the upstream transects, with a mean value of 11.79% in wet
season and 3.72% in dry season in the riparian areas. The mean SMC10 in the hillslopes was
6.58% in wet season and 2.86% in dry season. SMC20 showed similar fluctuation, 7.22% in wet
season and 2.98% in dry season in the riparian areas and 7.56% in wet season and 4.4% in dry
season in the hillslopes. In transect T5, average SMC10 and SMC20 at the center of the lake
(29.00% in wet season and 13.36% in dry season; 29.30% in wet season and 9.69% in dry season)
were higher than those along the lake shore (4.90% in wet season and 3.13% in dry season; 3.34%
in wet season and 5.22% in dry season).

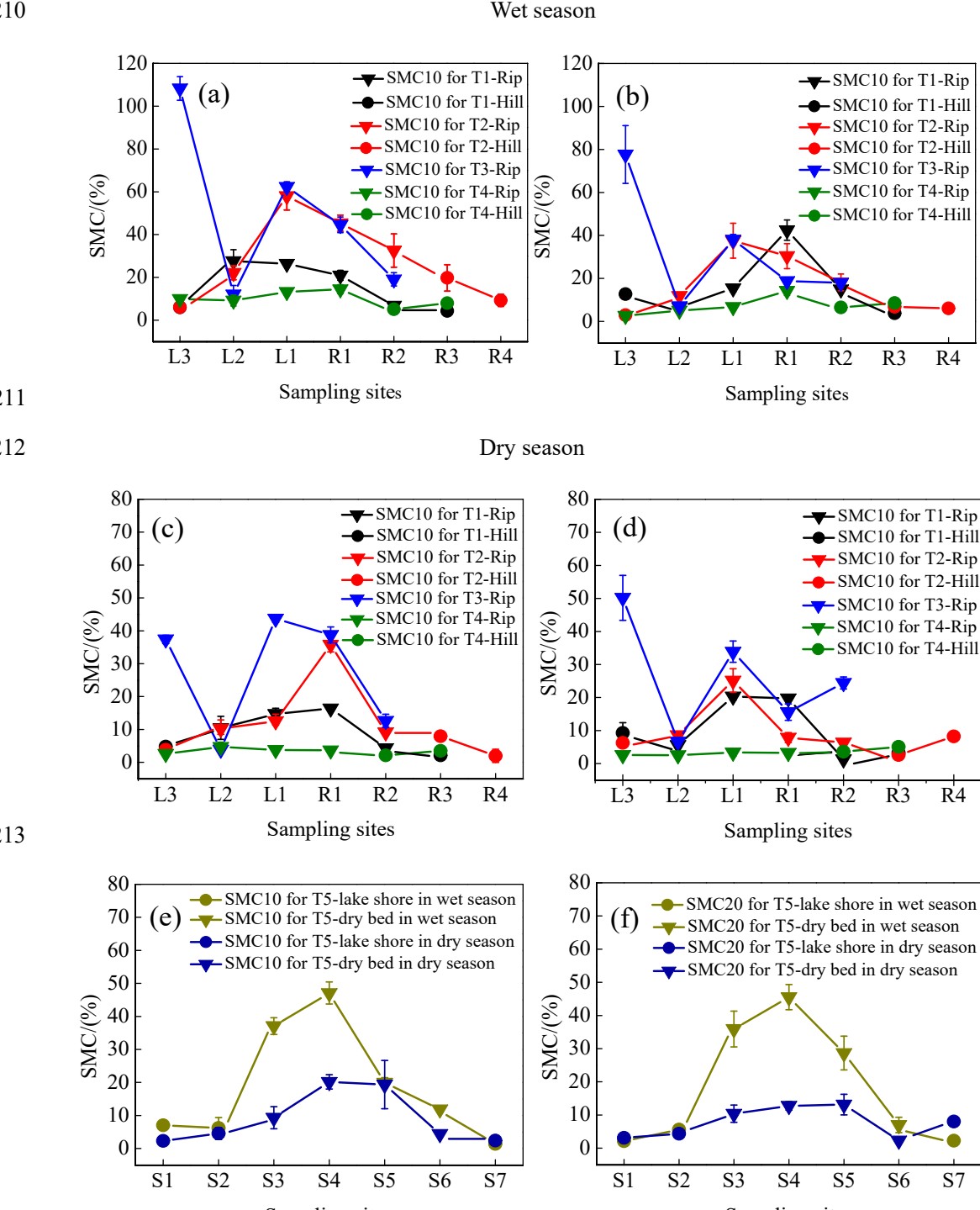



Fig. 3 Soil mass moisture contents (SMCs) at soil depths of 0–10 cm (SMC10) and 10–20 cm
(SMC20) for transects T1–T5 in wet season and dry season. Error bars represent the SD about the
mean.

**3.2  Spatiotemporal patterns of ST in each transect**
Spatiotemporal differences in ST during the entire observation period are displayed in Fig. 4.
ST variations in wet season (mean value: 27.4°C) were noticeably higher than those in dry season
(mean value: 8.97°C). Moreover, ST for riparian sites (mean values: 26.0°C in wet season and
8.41°C in dry season) was slightly lower than that for the hillslope grasslands (mean values:
30.9°C in wet season and 10.3°C in dry season) for the 0–10 cm soil depth, with the exception of
transect T5. Similar results were observed for the 10–20 cm soil depth.
Wet season

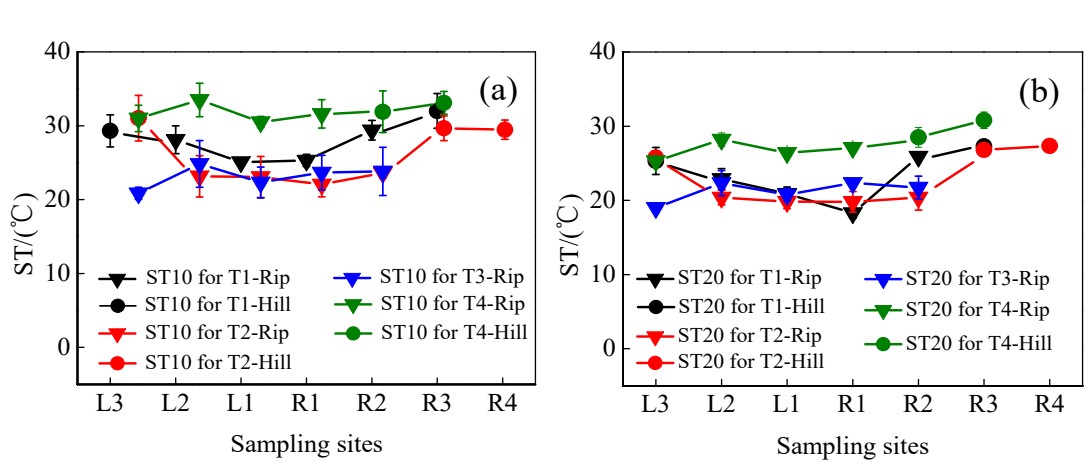


Dry season

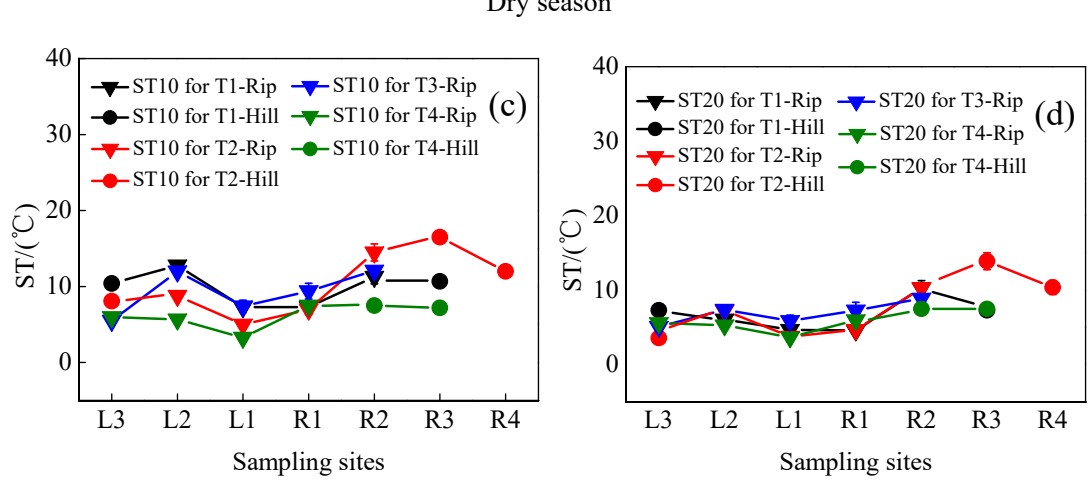


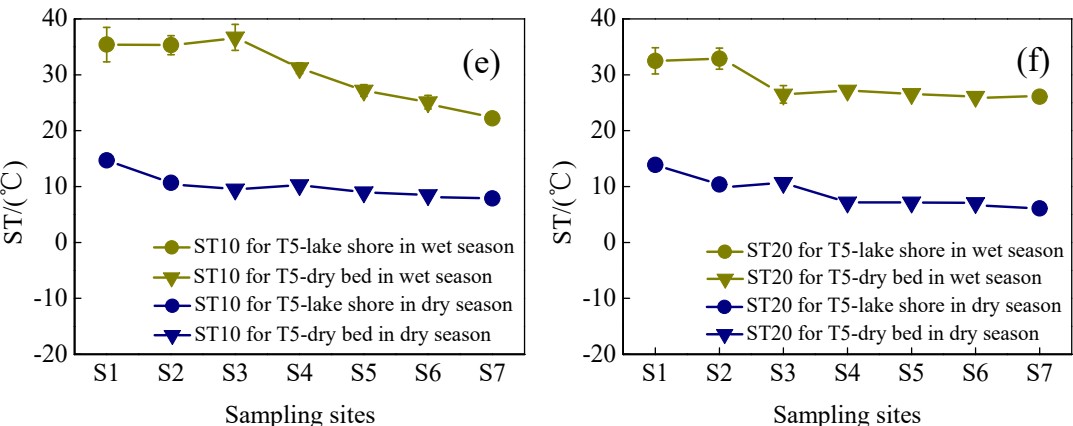


Fig. 4 Soil temperatures (STs) at soil depths of 0–10 cm (ST10) and 10–20 cm (ST20) for
transects T1–T5 in wet season and dry season. Error bars represent the SD about the mean.

## 3.3 Spatiotemporal patterns of GHG emissions in each transect

Figure 5 shows the spatiotemporal variations in GHG emissions in wet season and dry season
in each transect. $CO_2$ emissions in each transect were higher in wet season than in dry season. The
average emissions for the riparian wetlands of transects T1–T4 (1582.09 ± 679.34 mg·m$^{-2}$·h$^{-1}$ in
wet season and 163.24 ± 84.98 mg·m$^{-2}$·h$^{-1}$ in dry season) were higher than those for the hillslope
grasslands (1071.54 ± 225.39 mg·m$^{-2}$·h$^{-1}$ in wet season and 77.68 ± 25.32 mg·m$^{-2}$·h$^{-1}$ in dry
season). Higher $CO_2$ fluxes occurred in the riparian zones, while lower $CO_2$ fluxes were observed
in the hillslope grasslands in continuous river transects (T1, T2, and T3). Transect T4 exhibited
lower $CO_2$ emissions in the riparian wetlands near the channel than at sites away from the channel.
$CO_2$ emissions in transect T5 in wet season and dry season decreased from the lake shore to the
lake center.

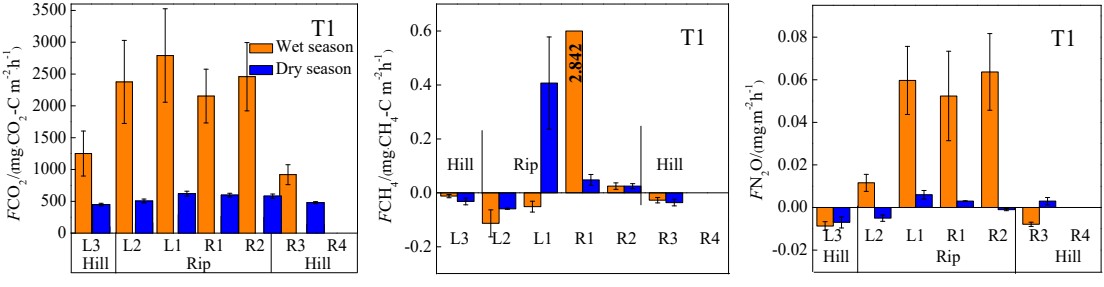


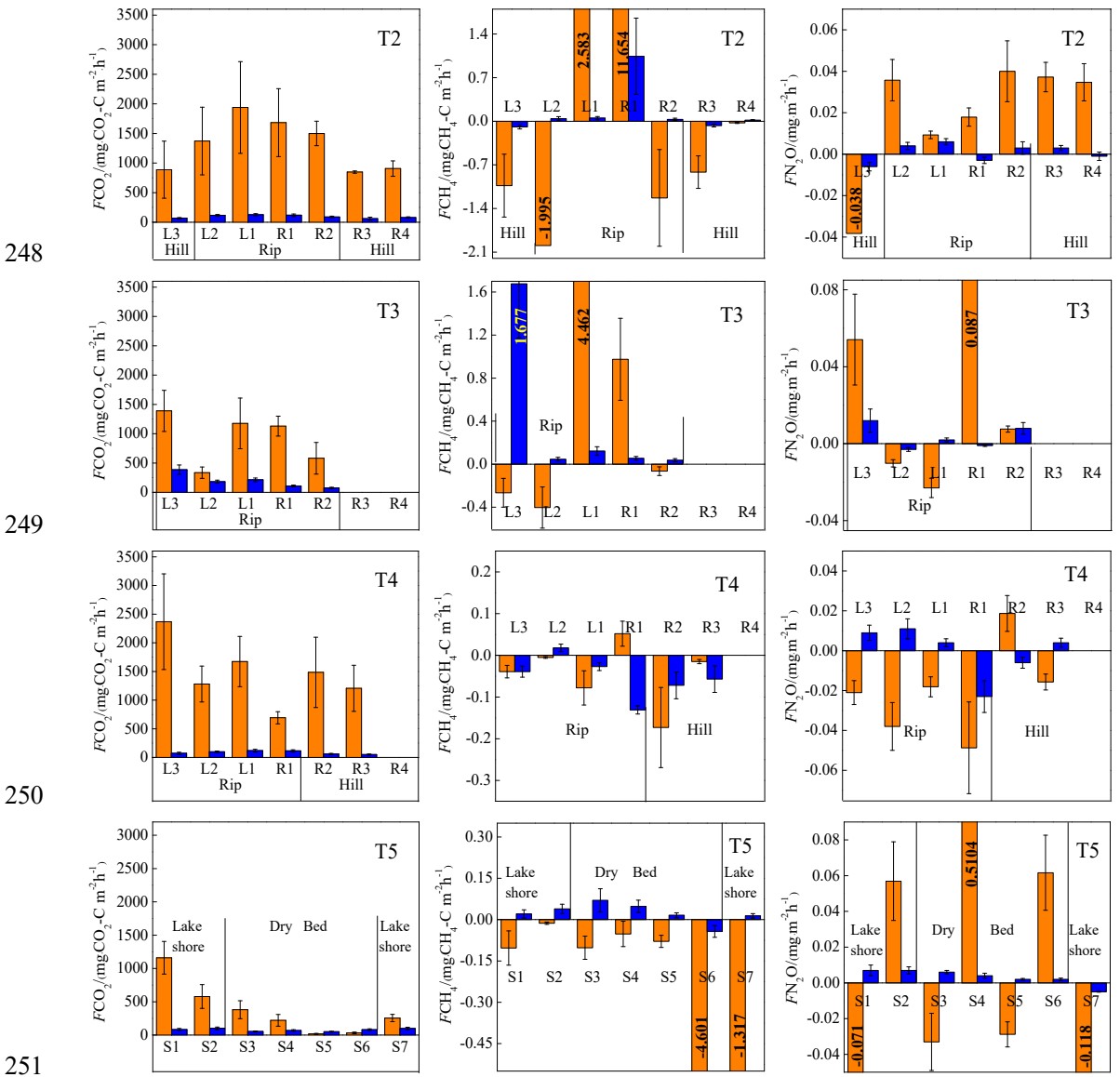





Fig. 5 Spatiotemporal patterns of $CO_2$ (first column), $CH_4$ (second column), and $N_2O$
(third column) emissions (*F*) for each transect. Data are shown for wet season (orange) and dry
season (blue) and error bars are the standard deviations.

$CH_4$ emissions at the transects with continuous river flow (T1, T2, and T3) varied between
wet season and dry season, except for T4 (characterized by intermittent river flow) and T5 (the dry
lake). In wet season, the near-stream sites (L1 and R1) in T1, T2, and T3 were characterized as
high $CH_4$ sources (average: $3.74 \pm 3.81$ mg·m$^{-2}$·h$^{-1}$), but the sites located away from the river
gradually turned into $CH_4$ sinks. Moreover, all the sites in transects T4 and T5 were sinks. $CH_4$
emissions (mean value: $0.2 \pm 0.45$ mg·m$^{-2}$·h$^{-1}$) at the wetland sites were always lower in dry
season than those in wet season. However, the sites on the hillslope grasslands served as $CH_4$
sinks (mean value: $-0.05 \pm 0.03$ mg·m$^{-2}$·h$^{-1}$). In transect T5, CH$_4$ emissions revealed the opposite
trend; a CH$_4$ sink was observed in wet season, but it was transformed into a CH$_4$ source in dry
season.

Similar to the CO$_2$ and CH$_4$ emissions, N$_2$O emissions showed a distinct spatiotemporal

pattern for all the transects. N$_2$O emissions in wet season were higher than those in dry season.
These emissions were higher in riparian wetlands than in hillslope grasslands. Moreover, almost
all the sites with continuous river flow were N$_2$O sources, while more than half of the sites with
intermittent river flow were sinks.

Table 3 shows that CO$_2$ fluxes were significantly correlated between the wet season and dry

season, while CH$_4$ and N$_2$O fluxes were not correlated in two seasons.

Table 3 Significant correlations between GHGs fluxes and two seasons (n-31)

| GHG flux | FCO$_2$ in wet season-FCO$_2$ in dry season | FCH$_4$ in wet season-FCH$_4$ in dry season | FN$_2$O in wet season- FN$_2$O in dry season |
|---|---|---|---|
| significant correlations (P) | 0.000 | 0.133 | 0.290 |

Note: P<0.05 denote significant correlations and P > 0.05 denote no significant correlations
**3.4 Spatiotemporal patterns of GHG emissions in upstream and downstream**
**areas**

Figure 6 shows the detailed spatial and seasonal distribution of GHG emissions in wet season

and dry season in the longitudinal direction from the upstream (T1, T2, and T3) to the downstream
areas (T4 and T5). The CO$_2$, CH$_4$, and N$_2$O emissions were calculated from the average values of
the respective emissions in the wetlands and hillslope grasslands in each transect.

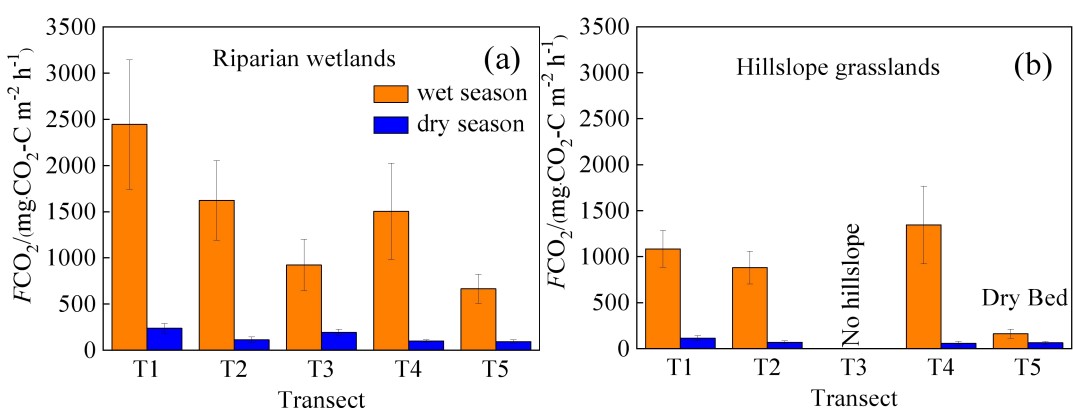


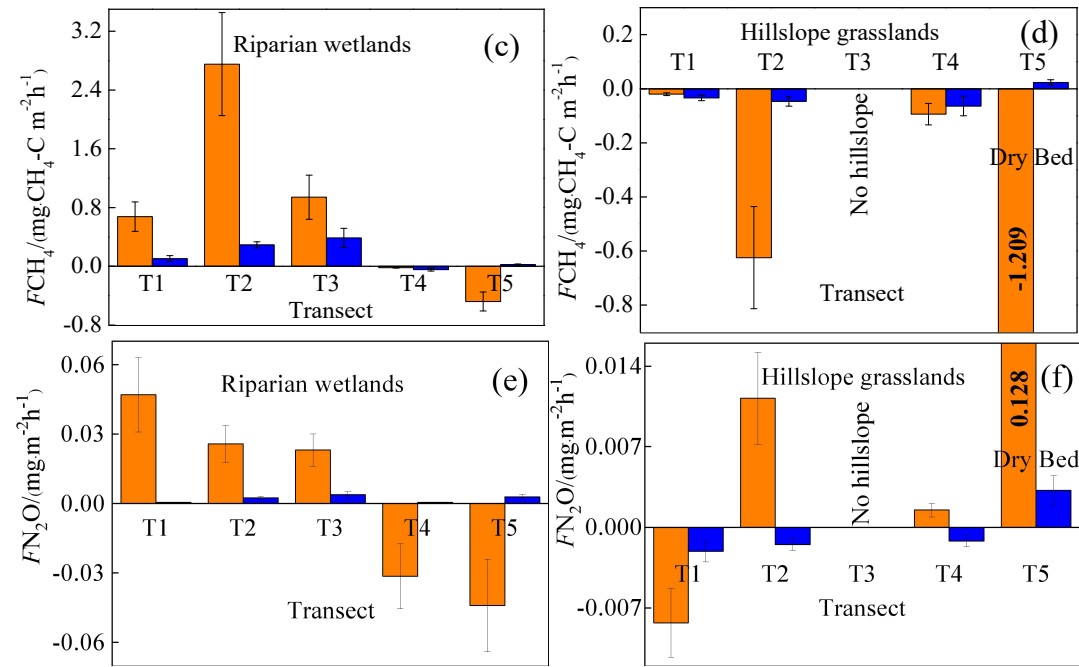



Fig. 6 Spatiotemporal patterns of $CO_2$ (first line), $CH_4$ (second line), and $N_2O$ (third line)

emissions ($F$) in the upstream (T1, T2, and T3) and downstream areas (T4 and T5). Bars are the

mean values for each transect and error bars are the standard errors.

$CO_2$ emissions in riparian wetlands (Fig. 6(a)) in wet season decreased from $2444.69 \pm$ $228.58$ mg·m$^{-2}$·h$^{-1}$ in the upstream area to $665.08 \pm 347.57$ mg·m$^{-2}$·h$^{-1}$ downstream, and the corresponding values for dry season were $238.12 \pm 48.20$ mg·m$^{-2}$·h$^{-1}$ and $94.14 \pm 7.67$ mg·m$^{-2}$·h$^{-1}$. However, in hillslope grasslands (Fig. 6(b)), $CO_2$ emissions exhibited no significant seasonality between upstream and downstream areas, with the mean values of $1103.40 \pm 190.44$ mg·m$^{-2}$·h$^{-1}$ in wet season and $79.18 \pm 24.52$ mg·m$^{-2}$·h$^{-1}$ in dry season. In addition, $CO_2$ emissions in transect T5 were lower for both months, with the averages of $162.83 \pm 149.15$ mg·m$^{-2}$·h$^{-1}$ and $63.26 \pm 12.40$ mg·m$^{-2}$·h$^{-1}$ in wet season and dry season, respectively. The upstream riparian zones exhibited higher $CO_2$ emissions ($894.32 \pm 868.47$ mg·m$^{-2}$·h$^{-1}$) than their downstream counterparts ($621.14 \pm 704.10$ mg·m$^{-2}$·h$^{-1}$). However, mean $CO_2$ emissions showed no significant differences in grasslands, averaging $524.16 \pm 450.10$ mg·m$^{-2}$·h$^{-1}$ upstream and $508.06 \pm 534.77$ mg·m$^{-2}$·h$^{-1}$ downstream.

$CH_4$ emissions showed a marked spatial pattern in the riparian zones from upstream to downstream (Fig. 6(c)). The transects with continuous river flow were $CH_4$ sources in wet season and dry season, with the average emissions of $1.42 \pm 3.41$ mg·m$^{-2}$·h$^{-1}$ and $0.27 \pm 0.49$ mg·m$^{-2}$·h$^{-1}$,

respectively, while those with intermittent river flow served as $CH_4$ sinks, with the corresponding
mean values of $-0.21 \pm 0.45$ mg·m$^{-2}$·h$^{-1}$ and $-0.02 \pm 0.05$ mg·m$^{-2}$·h$^{-1}$. Moreover, the hillslope
grassland sites in all transects were $CH_4$ sinks (Fig. 6(d)).

$N_2O$ emissions in riparian wetlands (Fig. 7(e)) showed spatial patterns similar to those of

$CH_4$ emissions. In wet season, the transects with continuous river flow served as $N_2O$ sources,
with the mean value of $0.031 \pm 0.031$ mg·m$^{-2}$·h$^{-1}$, while those with intermittent river flow were
$N_2O$ sinks with an average value of $-0.037 \pm 0.05$ mg·m$^{-2}$·h$^{-1}$. In dry season, $N_2O$ emissions
occurred as weak sources in the longitudinal transects, averaging $0.002 \pm 0.007$ mg·m$^{-2}$·h$^{-1}$.
However, $N_2O$ emissions in hillslope grasslands did not show any spatial pattern (Fig. 7(f)).
**4. Discussion**
**4.1 Main factors influencing GHG emissions**
4.1.1 Effects of SMC on GHG emissions

SMC constituted one of the main factors affecting GHG emissions in wetlands. In this study,

transects T1–T4 were characterized by a marked spatial SMC gradient (i.e., a gradual decrease
include SMC10 and SMC20 from the riparian wetlands to the hillslope grasslands and from
upstream to downstream (Fig. 3)). The $CO_2$, $CH_4$, and $N_2O$ emissions showed a similar trend. In
Table 4, SMC10 is positive correlated with $CO_2$ emissions ($P < 0.05$), SMC10 and SMC20 are
significantly positive correlated with $CH_4$ emissions ($P < 0.01$), and SMC10 and SMC20 are
highly positive correlated with $N_2O$ emissions ($P < 0.05$ and $P < 0.01$, respectively). These results
indicated the influence of wetland SMC on GHG emissions.

Typically, the optimal SMC values associated with $CO_2$ emissions in riparian wetlands range

from 40 to 60% (Sjögersten et al., 2006), creating better soil aeration and improving soil
microorganisms' activity and the respiration of plant roots, thereby promoting $CO_2$ emissions,
whereas excessive SMC reduces soil gas transfer due to the formation of an anaerobic
environment in the soil, and microbial activity is lower, favoring the accumulation of organic
matter (Hui., 2014). On the contrary, the SMC of hillslope grasslands is less than 10%. Low soil
moisture inhibits the growth of vegetation with few vegetation residues and litters. Meanwhile,
low soil moisture is not conducive to the survival of soil microorganisms, leading to a decrease in
$CO_2$ emissions than to those in riparian zones (Moldrup et al., 2000; Hui., 2014). Similar results
were obtained in our study. The changes in $CO_2$ emissions in transect T5 were contrary to the
change in the SMC10 and SMC20 likely because the optimal range of soil C:N is between 10-12
(Pierzynski et al., 1994), but the value in the dry lake bed of T5 is higher than 60, high soil C:N
resulted in nitrogen limitation in the process of decomposition of organic matter by
microorganisms. Furthermore, other sediment properties (like Soil pH>9.5) for this transect were not
conducive to the survival of microorganisms (Table 1), and the increase in SMC did not increase
the respiration activity of microorganisms.
The largest $CH_4$ emissions were observed at the near-stream sites (i.e., L1 and R1) in T1, T2,
and T3, with the average SMC of 30.29%, while the SMC values at the other sites, which were
either weak sources or sinks, averaged at 14.57%. These results indicate that a higher SMC is
favorable for $CH_4$ emissions because a higher SMC denotes a soil in a reduced state, which is
beneficial for $CH_4$ production and inhibits $CH_4$ oxidation. A similar result was reported by Xu et al.
(2008). They conducted experiments of $CH_4$ emissions from a variety of paddy soils in China, and
showed that $CH_4$ production rates increased with the increase in SMC at the same incubation
temperature. Meng et al. (2001) also reported that water depth was the main factor affecting $CH_4$
emissions from wetlands. When the water level dropped below the soil surface, the decomposition
of organic matter accelerated, and $CH_4$ emissions decreased. If the oxide layer is large, the soil is
transformed into a $CH_4$ sink (Meng net al., 2011).
The $N_2O$ fluxes showed a clear spatial pattern associated with the changes in SMC. The
moisture content of wetland soils directly affects the aeration status of the soil. Besides, the
aeration status affects the partial pressure of oxygen, which has an important impact on
nitrifying/denitrifying bacteria's activity and ultimately affects soil $N_2O$ emissions (Zhang et al.,
2005). Table 4 shows that $N_2O$ emissions are significantly positively correlated with SMC10 and
SMC20 ($P < 0.01$). Generally, when SMC was below the saturated water content, the
microorganisms were in an aerobic environment, and $N_2O$ mainly came from the nitrification
reaction. $N_2O$ emissions increases with the increase of SMC (Niu et al., 2017; Yu et al., 2006). In
our study, the sampling sites with higher SMC (riparian zones and some hillslope grassland zones
in the upstream transects) have higher $N_2O$ emissions. When SMC increases to the saturated water
content or is in a flooded state, the system was an anaerobic environment, and the Nos activity
was higher due to excessively high SMC, which was conducive to denitrification and eventually
produced $N_2$ (Niu et al., 2017; Yu et al., 2006), such as site L1 in transect T3 in this study. Ulrike
et al. (2004) showed that denitrification was the main process under flooded soil conditions in
wetland soils, and the release of $N_2$ exceeds $N_2O$. These findings are consistent with those of Liu
et al. (2003), who showed that SMC is an essential factor affecting $N_2O$ emissions.

Nitrification:

$$NH_4^+ \xrightarrow[\text{Nir}]{\text{AMO}} NH_2OH \longrightarrow [NOH] \xrightarrow{\text{HAO}} NO_2^- \xrightarrow{\text{NXR}} NO_3^-$$
$$\downarrow \qquad\qquad \downarrow$$
$$N_2O \xleftarrow{\text{Nor}} NO$$


Denitrification:

$$NO_3^- \xrightarrow{\text{Nar}} NO_2^- \xrightarrow{\text{Nir}} NO \xrightarrow{\text{Nor}} N_2O \xrightarrow{\text{Nos}} N_2$$


The enzymes involved in the formula include Ammonia monooxygenase (AMO),
Hydroxylamine oxidase (HAO), Nitrite REDOX enzyme (HAO), nitrate reductase (Nar), nitrite
reductase (Nir), Nitric oxide reductase (Nor) and Nitrous oxide reductase (Nos).
4.1.2 Effects of ST on GHG emissions
ST was another important factor affecting the $CO_2$ emissions in this study, as this parameter
was significantly correlated with $CO_2$ emissions ($P < 0.01$) (Table 4). The activity of soil
microorganisms increases with rising soil temperatures, leading to increased respiration, and
consequently higher $CO_2$ emissions (Heilman et al., 1999). Previous studies reported that ST
partially controls seasonal $CO_2$ emission patterns (Inubushi et al., 2003). Therefore, $CO_2$
emissions in wet season were significantly higher than those in dry season in this study.
$CH_4$ emissions showed a clear seasonal pattern because high summer temperatures improve
the activity of both $CH_4$-producing and -oxidizing bacteria (Ding et al., 2010). However, Table 4
indicates that the correlation between $CH_4$ emissions and temperature is not significant because
SMC could be more critical than temperature in our study region with very dry climate. SMC
showed a positive correlation with GHG emissions. In addition, SMC affected ST to a certain
extent, while the interactions between SMC and ST had a mutual influence on $CH_4$ emissions.
During the study period, the near-stream sites (L1 and R1) maintained a super-wet state on the
ground surface for a long time, which was beneficial for the production of $CH_4$. However, the
wetlands maintained a state without water accumulation on the soil surface in August, which was
conducive to the oxidative absorption of $CH_4$. SMC thus masked the effect of ST on $CH_4$
emissions.

Previous studies indicated that temperature is an important factor affecting $N_2O$ emissions

(Sun et al., 2011) through primary mechanisms impacting the nitrifying and denitrifying bacteria
in the soil. Table 4 shows that the correlations between $N_2O$ emissions and ST10 and ST20 are
poor ($P > 0.05$). This can be attributed to the wide suitable temperature range for
nitrification-denitrification and weak sensitivity to temperature. Malhi et al. (1982) found that the
optimum temperature for nitrification was 20 ℃, and it will inhibit entirely at 30 ℃. However,
Brady (1999) believed that the suitable temperature range for nitrification was 25～35℃, and the
nitrification inhibits below 5 ℃ or above 50 ℃. It showed that the temperature requirements of
nitrifying microorganisms in wetland soils were different in different temperature belts. The
suitable temperature range was the performance of the long-term adaptability of nitrifying
microorganisms. Meanwhile, several studies revealed that denitrification could be carried out in a
wide temperature range (5 ～ 70 ℃), and it was positively related to temperature (Fan., 1995).
However, the process will be inhibited when the temperature is too high or too low. The average
ST in wet season was 27.4℃, conducive to the growth of denitrifying microorganisms, while that
in dry season was 8.97℃, and the microbial activity was generally low (Sun et al., 2011).
Furthermore, ST fluctuations were low both in wet season and dry season. Therefore, the effect of
ST on $N_2O$ emissions was masked by other factors, such as moisture content.
4.1.3  Effects of BIO and soil organic matter on GHG emissions

$CO_2$ and $CH_4$ emissions were higher in the riparian wetlands than in the grasslands, mainly

because of greater vegetation cover. Typically, $CO_2$ emissions from riparian wetlands originate
from plants and microorganisms, with plant respiration accounting for a large proportion in the
growing season. Previous studies have shown that plant respiration accounts for 35–90% of the
total respiration in the wetland ecosystem (Johnson-Randall and Foote, 2005). Good soil
physicochemical properties and high soil total organic carbon (TOC) of riparian wetlands improve
the activity of soil microorganisms and plant root respiration. Table 4 shows that BIO is
significantly correlated with the $CO_2$ ($P < 0.05$) and $CH_4$ ($P < 0.01$) emissions. These results can
be attributed to the significant linear positive correlation between the respiration rate and plant
biomass (Lu et al., 2007). Higher plant biomass storage can achieve more carbon accumulation
during photosynthesis and higher exudate release by the roots. This, in turn, promotes the
accumulation of soil organic matter. Increased amount of organic matter stimulates the growth and
reproduction of soil microorganisms, ultimately promoting $CO_2$ and $CH_4$ emissions. Moreover,
plants act as a gas channel for $CH_4$ transmission, and a larger amount of biomass promotes $CH_4$
emissions, given the increased number of channels. In transect T3, high $CO_2$ emissions observed
at site L3 can be attributed to the relatively high levels of SMC, BIO, and soil nutrients, which
stimulate the microbial respiration rates.
BIO had a weak correlation with $N_2O$ emissions (Table 4), which indicates that plants
increase $N_2O$ production and emissions, although this may not be the most critical factor. Previous
studies reported mechanisms where in the plants can absorb $N_2O$ produced in the soil through the
root system before releasing it into the atmosphere. Additionally, the root exudates of plants can
enhance the activity of nitrifying and denitrifying bacteria in the soil, ultimately promoting the
production of $N_2O$. Finally, oxygen stress caused by plant respiration can regulate the production
and consumption of $N_2O$ in the soil, eventually affecting the conversion of nitrogen in the soil
(Koops et al., 1996; Azam et al., 2005).
Site L3 in transect T3 was covered by tall reeds, and its BIO was much higher than those of
the other sites; thus, the data for this site were excluded from the correlation analysis.

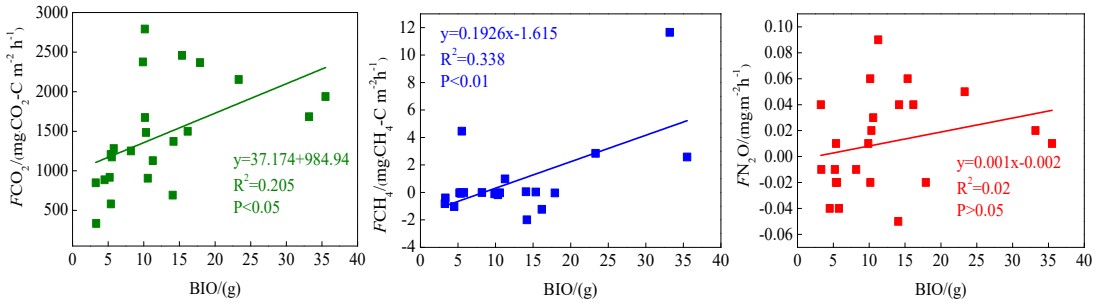

Fig. 7 Correlation between aboveground biomass (BIO) and GHG emissions ($F$)


Soil C:N ratio refers to the ratio of biodegradable carbonaceous organic matter and
nitrogenous matter in the soil, and it forms the soil matrix with TOC. TOC decomposition
provides energy for microbial activity, while the C:N ratio affects the decomposition of organic
matter by soil microorganisms (Gholz et al., 2010). The correlation results (Fig. 8) indicate that
TOC has a weak positive correlation with $CO_2$ emissions ($P > 0.05$), but soil C:N has a significant

negative correlation with $CO_2$ emissions ($P < 0.05$), indicating that nitrogen has a limiting effect on soil respiration by affecting microbial metabolism. Liu et al. (2019) reported that N addition promoted $CO_2$ emissions from wetlands soil, and the effect of organic N input was significantly higher than those of inorganic N input. Organic carbon provides a carbon source for the growth of plants and microorganisms, which boosts their respiration. Moreover, TOC has a significant correlation with $N_2O$ emissions ($P < 0.05$). Most heterotrophic microorganisms use soil organic matter as carbon and electron donors (Morley N and Baggs E M., 2010). Soil carbon source has an important influence on microbial activity. Nitrifying or denitrifying microorganisms need organic matter to provide carbon source during the assimilation of $NH_3$ or $NO_3^-$. The high content of organic matter in the soil can promote the abundance of heterotrophic nitrifying bacteria increases, consume dissolved oxygen in the medium, and cause the soil to become more anaerobic, slowing down autotrophic growth nitrifying bacteria. This reduces the nitrification rate, ultimately promoting $N_2O$ release. Enwall et al. (2005) studied the effect of long-term fertilization on soil denitrification microbial action intensity. They found that the soil with long-term organic fertilizer application has a significant increase in organic matter content, and consequently, a significant increase in denitrification activity. Typically, low soil C:N ratios are favorable for the decomposition of microorganisms, the most suitable range being between 10 and 12 (Pierzynski et al., 1994). Table 4 shows that $N_2O$ emissions are significantly related to the soil C:N ratios ($P < 0.05$), which means that denitrifying bacteria will use their endogenous carbon source for denitrification when the external carbon source is insufficient. Moreover, incomplete denitrification leads to the accumulation of $NO_2$-N, which is conducive to the $N_2O$ release. Meanwhile, due to the weak competitive ability of Nos to electrons, low C:N inhibits the synthesis of Nos, which is also a reason for $N_2O$ release. In this study, all the sites in transects T1–T4 exhibited similar soil C:N ratios in the optimum range (Table 1), which is favorable for microbial decomposition. However, the soil C:N ratios in transect T5 were higher than those in the other transects, especially in the dry lake bed. Therefore, transect T5 showed severe mineralization and a low microbial decomposition rate.

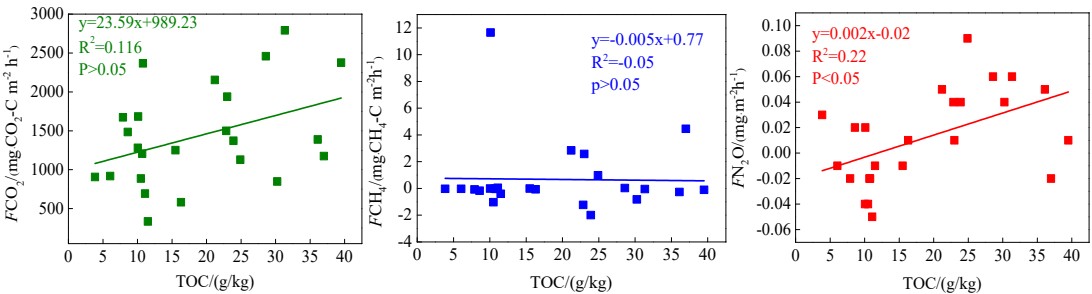


Fig. 8 Correlations between soil organic carbon (TOC) and GHG emissions ($F$)

Table 4. Correlations between $CO_2$, $CH_4$, and $N_2O$ emissions and impact factors ($n = 62$)

| GHG flux | ST10 | ST20 | SMC10 | SMC20 | TOC | $\rho_b$ | C:N | pH | EC | BIO |
|----------|------|------|-------|-------|-----|----------|-----|-----|-----|-----|
| $CO_2$ | 0.634** | 0.592** | 0.307* | 0.216 | 0.393 | −0.463** | −0.289* | −0.350** | −0.251* | 0.491* |
| $CH_4$ | −0.029 | −0.051 | 0.346** | 0.353** | −0.02 | −0.129 | −0.156 | −0.127 | −0.107 | 0.607** |
| $N_2O$ | 0.127 | 0.118 | 0.304* | 0.356** | 0.493* | −0.194 | 0.311* | 0.137 | 0.504** | 0.251 |

Note: 1. The analysis method used in the table is Pearson correlation analysis, and the numbers
represent Pearson correlation coefficients.
2. * and ** denote significant and highly significant correlations ($P < 0.01$ and $P < 0.05$),
respectively.
3.  ST - soil temperature, SMC - soil moisture content, $\rho_b$ - soil bulk density, soil C:N - soil
carbon-nitrogen ratio, pH - soil pH, EC - soil electrical conductivity, BIO - aboveground biomass

**4.2 Riparian wetlands as hotspots of GHG emissions**

The results of this study emphasized that $CO_2$ emissions in the riparian wetlands were higher

than those in the hillslope grasslands owing to a variety of factors. ST is an important factor
affecting GHG emissions. Mclain and Martens (2006) showed that seasonal fluctuations in ST and
SMC in semi-arid regions have important effects on $CO_2$, $CH_4$, and $N_2O$ emissions in riparian
soils. Poblador et al. (2017) studied the GHG emissions in forest riparian zones and suggested that
the difference in the $CO_2$ and $N_2O$ emissions in these zones is affected by the spatial gradient of
the regional SMC. In this study, the upstream riparian wetlands are characterized by higher TOC,
lower soil C:N ratio, and abundant BIO than the hillslope grasslands (Table 1). These soil
conditions benefited the soil microbial activity, ultimately enhancing respiration as well as $CO_2$
emissions. However, $CO_2$ emissions in downstream areas were nearly identical to those in the
grasslands because the wetlands gradually evolved into grasslands after their degradation. The
$N_2O$ emissions showed spatial patterns similar to those of the $CO_2$ emissions because the $CO_2$
concentrations were closely related to nitrification and denitrification processes. High $CO_2$
concentrations can promote the carbon and nitrogen cycles in soil (Azam et al., 2005), increasing
below ground C allocation associated with increased root biomass, root turnover, and root
exudation in elevated $pCO_2$ plants provided the energy for denitrification in the presence of high
available N, or that there was increased $O_2$ consumption under elevated $pCO_2$ (Baggs et al., 2003).
Moreover, soil respiration increases during soil denitrification (Liu et al., 2010; Christensen et al.,
1990). In this study, a weak correlation was observed between the $CO_2$ and $CH_4$ emissions in the
riparian zones ($r = 0.228$), but $CO_2$ emissions were significantly correlated with $N_2O$ emissions ($r$
$= 0.322$, $P < 0.05$). The soil became anaerobic in the riparian areas as the SMC increased, and this
was conducive to the survival of $CH_4$-producing bacteria and denitrification reactions, eventually
leading to an increase in $CH_4$ and $N_2O$ emissions. Jacinthe et al. (2015) reported that inundated
grassland-dominated riparian wetlands were $CH_4$ sinks ($-1.08 \pm 0.22$ kg$\cdot CH_4$-C ha$^{-1}\cdot$yr$^{-1}$), and Lu
et al. (2015) also indicated that grasslands were $CH_4$ sinks. In our study, a marked water gradient
across the transects led to the transformation of the soil from anaerobic to aerobic soil, which
changed the wetland function as a $CH_4$ source or sink. Therefore, during the transition from the
riparian wetlands to the hillslope grasslands, $CH_4$ emissions only appeared as sources in the
near-stream sites and sinks at other sites.

Further, we compared the GHG emissions of riparian wetlands and hillslope grasslands

around the Xilin River Basin with various types of grasslands (meadow grassland, typical
grassland, and desert grassland) in the Xinlingol League in Inner Mongolia (Table 5). The $CO_2$
emissions in wet season decreased in the following order: upstream riparian wetlands >
downstream riparian wetlands > hillslope grasslands > meadow grassland > typical grassland >
desert grassland. Moreover, the upper riparian wetlands acted as source of $CH_4$ emissions, while
the downstream transects and grasslands served as $CH_4$ sinks. Similarly, except for the
downstream transects, $N_2O$ emissions occurred as weak sources in different types of grasslands
and upstream riparian wetlands. The GHG emissions showed similar spatial patterns in October.
Although these estimates were made only in the growing season in August and the non-growing
season in October, our results suggest that riparian wetlands are the potential hotspots of GHG
emissions. Thus, it is important to study GHG emissions to obtain a comprehensive picture of the

 role of riparian wetlands in climate change.


Table 5. GHG emission fluxes of riparian wetlands and grasslands

| Sample plot | | GHG emissions in August (mg·m$^{-2}$·h$^{-1}$) | | | GHG emissions in October (mg·m$^{-2}$·h$^{-1}$) | | | Reference |
|---|---|---|---|---|---|---|---|---|
| | | $CO_2$ | $CH_4$ | $N_2O$ | $CO_2$ | $CH_4$ | $N_2O$ | |
| Wetlands of upstream transects (T1, T2, and T3) | n=13 | 1606.28 ± 697.78 | 1.417 ± 3.41 | 0.031 ± 0.03 | 182.35 ± 88.26 | 0.272 ± 0.49 | 0.002 ± 0.005 | This study |
| Wetlands of downstream transects (T4 and T5) | n=7 | 1144.15 ± 666.50 | −0.215 ± 0.45 | −0.037 ± 0.05 | 98.13 ± 15.11 | −0.015 ± 0.05 | 0.001 ± 0.01 | |
| Hillslope grasslands of all transects | n=7 | 1071.54 ± 225.39 | −0.300 ± 0.40 | 0.003 ± 0.03 | 77.68 ± 25.32 | −0.048 ± 0.03 | −0.002 ± 0.005 | |
| Meadow grassland | | 166.39 ± 45.89 | −0.038 ± 0.009 | 0.002 ± 0.001 | - | - | - | |
| Typical grassland | | 240.32 ± 87.56 | −0.042 ± 0.025 | 0.037 ± 0.034 | - | - | - | Guo et al., 2017 |
| Desert grassland | | 107.59 ± 54.10 | −0.036 ± 0.015 | 0.003 ± 0.001 | - | - | - | |
| Typical grassland | | 520.25 ± 59.07 | −0.102 ± 0.012 | 0.007 ± 0.001 | 88.34 ± 9.84 | −0.099 ± 0.003 | 0.005 ± 0.001 | Zhang, 2019 |
| Typical grassland | | 232.42 ± 18.90 | −0.090 ± 0.005 | 0.004 ± 0.001 | - | - | - | Chao, 2019 |
| Typical grassland | | 265.23 ± 31.43 | −0.185 ± 0.018 | 0.005 ± 0.001 | 189.41 ± 28.96 | −0.092 ± 0.012 | 0.004 ± 0.001 | |
| Meadow grassland | | 553.85 | −0.163 | 0.003 | 47.73 | −0.019 | 0.011 | Geng, 2004 |
| Typical grassland | | 308.60 | −0.105 | 0.002 | 70.25 | −0.029 | 0.007 | |


We roughly estimated the annual cumulative emissions of $CO_2$, $CH_4$, and $N_2O$ from riparian
wetlands and hillslope grasslands around the Xilin River Basin, and further calculated its global
warming potential. Table 6 indicated that annual cumulative emissions of $CO_2$ and $CH_4$ decreased
in the following order: upstream riparian wetlands > downstream riparian wetlands > hillslope
grasslands, and $N_2O$ in the following order: upstream riparian wetlands > hillslope grasslands >
downstream riparian wetlands. In this study, we used the static dark-box method to measure $CO_2$

emissions, which does not consider the absorption and fixation of $CO_2$ by plants' photosynthesis. Therefore, the total annual cumulative $CO_2$ emissions are high. This result clearly showed that the significant impact of $CO_2$ emissions than $CH_4$ and $N_2O$ emissions on global warming. The GWP depends on the cumulative emissions of the GHGs. GWP is shown as (Table 6): upstream riparian wetlands (13474.91 kg/hm$^2$) > downstream riparian wetlands (8974.12 kg/hm$^2$) > hillslope grasslands (8351.24 kg/hm$^2$). Therefore, both riparian wetlands and grasslands are the "sources" of GHGs on a 100-year time scale. The source strength of wetlands is higher than grasslands, further indicating that riparian wetlands are the hotspots of GHG emissions.

Table 6 Cumulative  annual emission flux and global warming potential of GHGs in riparian wetlands and grasslands

| Sample plot | $CO_2$/kg/hm$^2$ | $CH_4$/kg/hm$^2$ | $N_2O$/kg/hm$^2$ | GWP/$CO_2$ kg hm$^2$ |
| --- | --- | --- | --- | --- |
| Wetlands of upstream transects (T1, T2, and T3) | 13092.8±5378.16 | 12.36±26.40 | 0.25±0.23 | 13474.91±5828.68 |
| Wetlands of downstream transects (T4 and T5) | 9093.47±4831.82 | -1.68±3.23 | -0.26±0.40 | 8974.12±4912.75 |
| Hillslope grasslands of all transects | 8412.26±1614.26 | -2.55±3.12 | 0.01±0.20 | 8351.24±1648.22 |

**4.3 Effects of riparian wetland degradation on GHG emissions**

The hydrology and soil properties showed evident differences among the transects because the downstream zone was dry all year due to the presence of the Xilinhot Dam (Fig. 1). The dam caused the degradation of the riparian wetlands, resulting in reduced GHG emissions. The average $CO_2$ emissions amounted to 1663 mg·m$^{-2}$·h$^{-1}$ in the riparian wetlands in the upstream transects (T1, T2, and T3), while the downstream transects (T4 and T5) recorded an average of 1084 mg·m$^{-2}$·h$^{-1}$, 35% lower than the value in the upstream transects. The $N_2O$ emissions from the riparian wetlands were lower in the downstream transects.

The wetland degradation first resulted in the continuous reduction of SMC, which led to the deepening of the wetland's aerobic layer thickness. Besides, SMC could affect ST's change and thus transformed $CH_4$ emissions from a source to a sink by affecting methanogens' activity (Yan et al., 2018). Secondly, the reduction of SMC impeded aboveground plants' physiological activities

and inhabited related enzymes' activities in the respiration process. Meanwhile, various enzyme reactions of underground microorganisms under water stress influence and reduced $CO_2$ emissions (Zhang et al., 2017). Finally, after wetland degradation, long-term drought caused too low SMC, which was not conducive to the growth of nitrifying and denitrifying bacteria, which caused the transformation of $N_2O$ emissions from source to sink (Zhu et al., 2013). Table 1 shows that soil TOC in the upstream transects (average: 25.1 g·kg$^{-1}$) is higher than that in the downstream transects (average: 8.41 g·kg$^{-1}$). The relatively low SMC and the aerobic environment were conducive to the mineralization and decomposition of TOC. The degradation of plants in the wetlands led to the gradual reduction of BIO. Ultimately, the plant carbon source input of the degraded wetlands decreased, and the bare land temperature increased due to the reduced plant shelter. This accelerated the decomposition of TOC, leading to its decrease. This result indicates that wetland degradation caused the soil carbon pool's loss and weakened the wetland carbon source/sink function. These results are in agreement with those of Xia (2017).

The degraded wetlands also caused soil desertification and salinization, leading to a decline in the physical protection afforded by organic carbon and a reduction in soil aggregates. Thus, the preservation provided by organic carbon declined. TOC and SMC in the dry lake bed in transect T5 were relatively high, but GHG emissions were very low along this transect because soil pH values increased after the degradation of the lake soil, exceeding the optimum range required for microorganism activity. The soil C:N ratio was very high, resulting in severe mineralization and a low microbial decomposition rate, hence affecting the GHG emissions.

## 5. Conclusions

The riparian wetlands in the Xilin River Basin constitute a dynamic ecosystem. The present spatial and temporal transfers in the studied biogeochemical processes were attributed to the changes in SMC, ST, and soil substrate availability. Our simultaneous analysis of $CO_2$, $CH_4$, and $N_2O$ emissions from riparian wetlands and hillslope grasslands in the Xilin River Basin revealed that the majority of the GHG emissions occurred in the form of $CO_2$. Moreover, our results clearly illustrated a marked seasonality and spatial pattern of GHG emissions along the transects and in the longitudinal direction (i.e., upstream and downstream). SMC and ST were two critical factors

controlling the GHG emissions. Moreover, abundant BIO promoted the $CO_2$, $CH_4$, and $N_2O$
emissions.
The riparian wetlands were the potential hotspots of GHG emissions in the Inner Mongolian
region. However, the degradation of wetlands transformed the area from a source to a sink for $CH_4$
and $N_2O$ emissions, and reduced $CO_2$ emissions, which severely affected the wetland carbon cycle
processes. Our results show that the riparian wetlands have high $CO_2$ emissions, but wetlands are
$CO_2$ sink in the overall $CO_2$ balance general due to the photosynthesis of plants. Overall, our study
suggests that anthropogenic activities have significantly changed the hydrological characteristics
of the studied area, and will accelerate carbon loss from the riparian wetlands and further
influence the GHG emissions in the future.

**Author Contributions**

Xinyu Liu, Xixi Lu and Ruihong Yu designed the research framework and wrote the
manuscript. Xixi Lu and Ruihong Yu supervised the study. Xinyu Liu, Hao Xue, Zhen Qi,
Zhengxu Cao and Zhuangzhuang Zhang carried out the field experiments and laboratory
experiments analyses. Z.Z. drew GIS mapping in this paper. Tingxi Liu proofread the manuscript.
Heyang Sun contributed much in the revised version of our manuscript.

**Acknowledgements**

This study was funded by the National Key Research and Development Program of China
(grant no. 2016YFC0500508), Major Science and Technology Projects of Inner Mongolia
Autonomous Region (grant nos. 2020ZD0009 and ZDZX2018054), National Natural Science
Foundation of China (grant no. 51869014), Key Scientific and Technological Project of Inner
Mongolia (grant no. 2019GG019), and Open Project Program of the Ministry of Education Key
Laboratory of Ecology and Resources Use of the Mongolian Plateau (grant no. KF2020006).

**Competing interests**

The authors declare no conflicts of interest.

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
