# Peer review of "Greenhouse gases emissions from riparian wetlands: An example from the Inner Mongolia grassland region in China"

_Biogeosciences, 2020_

## Referee Comment (RC1) · Anonymous Referee #1 · 22 Sep 2020

The paper is well written and has several interesting findings. The main conclusion is that riparian wetlands are potential hotspots of GHG emissions. However, some aspects have to be more elaborated and discussed. Please find the main critical points below.

a) Table 1: please provide the number of the samples (n). Moreover, the grain size distributions (or the %age of sand, silt, clay) should be added. Additionally, the saturated volumetric water content and the residual volumetric water content of the soil should be determined. b) Fig. 3: It is not clear if the SMC(%) is based on volume or mass. Also in the text the numbers for SMC are not clear. I suppose, the values are

[Figure]

gravitational SMCs. It is important that SMC is related to the soil water capacity and the pF curve of the soils. Therefore, relative saturation would be a better measure. Alternatively, the authors can define the field capacity and the saturation values of the different soils. c) Fig. 4) please integrate into the figures an improved legend. Then you can skip the lenghty text of fig.4. d) Fig. 6) please indicate Riparian wetlands and hillslope grasslands directly in the figures. Then you can shorten the lengthy text of fig. 6. e) line 292 and line 300/ line 301: SMC values of 40 to 60%... This must be related to the soil, because SMC is a function of suction (matrix potential). f) line 312: What means: "SMC was above the saturated water content"? This is not possible. g) Chapter 4.1.3: It would be beneficial for the understanding, if the authors can calculate Co2 balances. Is the balance of photosynthesis and respiration / emission positive or negative? h) The nitrification / denitrification description is too vague. Please insert the formulas of the nitrification / denitrification processes and determine its relation / quantification. i) table 3: please add the number of samples (n) j) line 464 and line 472: I would like to see the long term balance of Co2. Do we have a source or a sink in degraded wetlands considering a longer time span (several years)?

---

## Referee Comment (RC2) · Anonymous Referee #2 · 28 Jan 2021

General comments. This paper addresses the spatial (longitudinal and horizontal transects) and temporal (wet and dry) variability of CO2 CH4 and N2O fluxes in degraded and non-degraded riparian wetlands and compares them to the adjacent grassland ecosystems. Principal differences in GHG fluxes between different ecosystem types is related to different biogeochemical production and consumption processes as controlled by above ground biomass, soil moisture content, soil temperature, and soil carbon and nitrogen stocks. Overall, riparian wetlands sites were hotspots of GHG fluxes linked to higher soil moisture and soil carbon stocks than the adjacent grassland sites. However, the degraded wetlands shifted to minor sources of CO2 and sinks of CH4 and N2O as the soil moisture, carbon stocks, and redox conditions changed. This shift

may be significant in both local and regional GHG budgets when it comes to assessing the role of riparian wetlands to global warming and climate change. The paper may therefore be interesting for readers of biogeosciences as it offers some valuable details on the major controlling factors behind the shift in GHG source strengths between degraded and non-degraded riparian wetlands. However, some major flaws and information is missing in the M&M and results section that needs to be addressed to improve the manuscript quality (see specific comments). Statistical analysis should be more deepened and include a multiple variate analysis to explicitly show strengths of the individual controlling factors such as soil temperature and soil moisture content on GHG fluxes. The discussion needs to draw more on previous research by comparing similarities in findings to better support the discussion section. Particularly in terms of the biogeochemistry that may explain the spatial-temporal differences in the emissions from riparian wetlands.

Title Consider removing the word "river" as riparian wetlands already define wetlands on stream and riverbanks. Abstract Ln 13-15: Consider reversing the sentence to give details on the direct link between riparian wetlands and climate change. Introduction Ln 56: Remove the word "the" in "the nature. . .." Ln 79: Remove the word "the" in "at the local. . .." Materials and methods Ln 117: Replace "for" with "from" Ln 137: Consider replacing "the" with "a" in the reservoir bag. Ln 139: . . ..and or or for the sampling times. If "and", were they averaged for the day? It is a bit not clear now. Ln 141: . . ..oven-dried. Ln 148: Figure 2, colors of the site labels are too difficult to see, consider using more contrasting colors. Ln 155: Indicate whether they are means and SD or SE in table caption. Ln 166: Missing section on what statistical tests were used for the analysis of the results.

Results Ln 169: Variations in SMC? Ln 169- 171: Confusing, consider revising the sentence to make it clearer. Ln 173: Cite the section in figure 3 to enable the reader follow easily the results section. Ln 180: Consider indicating on the graphs season information to make it less confusing. i.e add wet on top of the first two graphs and

dry on the second pair of graphs. What are the error bars? Standard errors? Also throughout all the manuscript, consider using wet and dry instead of the months as it gives a more direct link to the hydrological conditions of the riparian wetlands. Ln 193: Not clearly seen in the graphs, maybe change the shapes of the points within the riparian region. Ln 196: Same comments as SMC on the visuals. LN 226: What statistical tests were used to show differences in the two seasons? This information is missing in the figure and in the text. Ln 247: Figure is stretched vertically. Check this for all figures to ensure the aspect ratio is maintained when adding them in the document.

Discussion Ln 282: The discussion includes results not shown in the results section. Consider shifting some of the results in the discussion to the results part of the manuscript. Ln 288: Indicate whether the correlation is positive or negative. Ln 292: Give more details on the mechanism that links SMC to CO2 fluxes that the authors found, and how it links with your findings. Ln 296: How was this shown in the results? Seems rather speculative. Possibly give ranges based on other studies and link them with your study as shown in Table 1. Ln 308: You mean aerobic decomposition. Ln 311: Is this shown in the results section? Not clear what value of SMC indicates the saturation water content. Ln 313: More details on how the Niu et al 2017 study relates to your study. Ln 316: What mechanism links increased SMC to higher N2O fluxes? Currently the information is missing. Ln 330: Confusing as you say its important at the start of the paragraph. Ln 336: Consider replacing the growing season to either August or October. Currently it is not clear which season is the growing season for a reader not familiar with the region of study. Ln 364: Do soil nutrients mean SOC. Not clear at the moment. Ln 380: remove "the" in "the soil C:N….." Ln 381: TOC is also part of the C:N ratio. Elaborate more on the disentanglement between the two in the point you are making. Ln 384: But the statistics show the correlation with TOC is not significant. Ln 389: Elaborate more how this promotes N2O release. Ln 403: More description required for the table. For example if the values given are correlation coefficients and the type of correlation test used. Ln 417: Table 1 also shows higher C:N ratios in riparian

soils. Ln 422: Elaborate more on the link between CO2 concentrations and nitrification denitrification processes to make it clearer for the reader. Ln 432: use "and" instead of "but" as the latter indicates differences in the findings of the two studies. Is that the case? If yes, consider reversing the sentence to clearly bring it out. Ln 442: remove "the" in as the sources of. . .. Ln 466: Was the soil carbon in the degraded wetlands lost through aerobic decomposition. Give more details on the mechanism. Conclusion Ln 486: Comparison of the source strengths of the three gases expressed as GWP not presented in the graphs. This may show more clearly that CO2 contributed more than the other two GHG. Consider adding it.

---

## Author Comment (AC1) · 1 Mar 2021

February 28, 2021

Dear Editors and Reviewersïij Ž

I very much appreciate your efforts and time in reviewing our manuscript. According to your precious advice and suggestions, we have revised this manuscript thoroughly. Response to each question from editors and reviewers were listed below. Thank you very much for your precious time and tremendous efforts in reviewing and supporting this manuscript.

Best Regards,

Xinyu Liu Inner Mongolia Key Lab of River and lake ecology & Ministry of Education Key Laboratory of Ecology and Resource Use of the Mongolian Plateau School of Ecology and Environment Inner Mongolia University Room 106, Biology Building No. 235, West University Road, Saihan District, Hohhot Inner Mongolia 010021, P. R. China Mobile: +86-13245131615 E-mail: 21815009@mail.imu.edu.cn

Reviewers' comments: Reviewer #1: a)Table 1: please provide the number of the samples (n). Moreover, the grain size distributions (or the % age of sand, silt, clay) should be added. Additionally, the saturated volumetric water content and the residual volumetric water content of the soil should be determined. Reply: We have added the number of the samples (n), annual soil volumetric moisture content for the 0–10 cm and 10–20 soil depth in wet season and in dry season, and the saturated soil moisture content (SSM) in Table 1 and the grain size distributions added in Table 2. However, we don't add the residual volumetric water content, because we cannot measure the matrix suction and draw the pF curve. Residual volumetric water usually obtained by fitting the pF curve with the van genuchten formula. This is another research direction, and we do not have enough theory to study it. b)Fig. 3: It is not clear if the SMC(%) is based on volume or mass. Also in the text the numbers for SMC are not clear. I suppose, the values are gravitational SMCs. It is important that SMC is related to the soil water capacity and the pF curve of the soils. Therefore, relative saturation would be a better measure. Alternatively, the authors can define the field capacity and the saturation values of the different soils. Reply: SMC stands for soil mass moisture content, which has been indicated on line 144. We have rewritten the contents of the SMC, marking SMC10 and SMC20 as following: "The temporal and spatial variations in SMC10 in the following order: wet season > dry season and riparian wetlands > hill-slope grasslands (Fig. 3a, c, e). Similar variations were observed in SMC20 (Fig. 3b, d, f). The average SMC10 and SMC20 in the continuous river transects in the riparian zones (37.44% in wet season and 19.40% in dry season; 25.96% in wet season and

17.39% in dry season) were higher than those in the hillslope grasslands (9.12% in wet season and 4.15% in dry season; 6.51% in wet season and 5.96% in dry season). During the study period, both SMC10 and SMC20 changed as the distance from the river increased, and the highest value was observed at the near-stream sites (L1 and R1). SMC10 fluctuations were low in the intermittent transect compared to the upstream transects, with a mean value of 11.79% in wet season and 3.72% in dry season in the riparian areas. The mean SMC10 in the hillslopes was 6.58% in wet season and 2.86% in dry season. SMC20 showed similar fluctuation, 7.22% in wet season and 2.98% in dry season in the riparian areas and 7.56% in wet season and 4.4% in dry season in the hillslopes. In transect T5, average SMC10 and SMC20 at the center of the lake (29.00% in wet season and 13.36% in dry season; 29.30% in wet season and 9.69% in dry season) were higher than those along the lake shore (4.90% in wet season and 3.13% in dry season; 3.34% in wet season and 5.22% in dry season)". c)Fig. 4) please integrate into the figures an improved legend. Then you can skip the lenghty text of fig.4. Reply: We have revised the legend in fig.4 and shortened the lengthy text of fig.4. d)Fig. 6) please indicate Riparian wetlands and hillslope grasslands directly in the figures. Then you can shorten the lengthy text of fig. 6. Reply: We have indicated "Riparian wetlands" and "Hillslope grasslands" in fig.6 and shortened the lengthy text of fig.6. e)line 292 and line 300/ line 301: SMC values of 40 to 60%... This must be related to the soil, because SMC is a function of suction (matrix potential). Reply: Yes, this is a very complex subject, and the soil's permeability is difficult to determine. This is another research direction, and we do not have enough theory to study it. So, we determined soil mass moisture content simply using experimental methods to illustrate the relationship between SMC and GHGs emissions. f)line 312: What means: "SMC was above the saturated water content"? This is not possible. Reply: Sorry for confusing you. "SMC was above the saturated water content" means that the soil reaches saturation. Thus, we have revised the sentence to "When SMC reaches or is close to saturation", which has been indicated on line 344. g)Chapter 4.1.3: It would be beneficial for the understanding, if the

authors can calculate CO2 balances. Is the balance of photosynthesis and respiration / emission positive or negative? Reply: The paper uses the static dark chamber method to measure the ecosystem's respiration and discusses the "emission" part of greenhouse gases. The "absorption" is not measured, so the CO2 balance cannot be calculated. This is a very good suggestion that can be studied in the future. Generally, photosynthesis in healthy wetlands is more significant than respiration, conducive to the accumulation of organic matter. During the wetlands' degradation, the plant community and microbial composition change, the biomass is reduced, and photosynthesis is minor than respiration, causing carbon loss in the wetlands. After wetlands completely degraded, photosynthesis is more excellent than respiration, reaching a new balance. However, compared with a healthy wetland, the accumulation of organic matter is significantly reduced. h)The nitrification / denitrification description is too vague. Please insert the formulas of the nitrification / denitrification processes and determine its relation / quantification. Reply: We have added the formula and modified it in various parts of 4.1.1, 4.1.2, and 4.1.3. "The N2O fluxes showed a clear spatial pattern associated with the changes in SMC. The moisture content of wetland soils directly affects the aeration status of the soil. Besides, the aeration status affects the partial pressure of oxygen, which has an important impact on nitrifying/denitrifying bacteria's activity and ultimately affects soil N2O emissions (Zhang et al., 2005). Table 4 shows that N2O emissions are significantly positively correlated with SMC10 and SMC20 (P < 0.01). Generally, when SMC was below the saturated water content, the microorganisms were in an aerobic environment, and N2O mainly came from the nitrification reaction. N2O emissions increases with the increase of SMC (Niu et al., 2017; Yu et al., 2006). In our study, the sampling sites with higher SMC (riparian zones and some hillslope grassland zones in the upstream transects) have higher N2O emissions. When SMC increases to the saturated water content or is in a flooded state, the system was an anaerobic environment, and the Nos activity was higher due to excessively high SMC, which was conducive to denitrification and eventually produced N2 (Niu et al., 2017; Yu et al., 2006), such as site L1 in transect T3 in this

study. Ulrike et al. (2004) showed that denitrification was the main process under flooded soil conditions in wetland soils, and the release of N2 exceeds N2O. These findings are consistent with those of Liu et al. (2003), who showed that SMC is an essential factor affecting N2O emissions". We have put the formula in the supplement. "Previous studies indicated that temperature is an important factor affecting N2O emissions (Sun et al., 2011) through primary mechanisms impacting the nitrifying and denitrifying bacteria in the soil. Table 4 shows that the correlations between N2O emissions and ST10 and ST20 are poor (P > 0.05). This can be attributed to the wide suitable temperature range for nitrification-denitrification and weak sensitivity to temperature. Malhi et al. (1982) found that the optimum temperature for nitrification was 20 ℃, and it will inhibit entirely at 30 ℃. However, Brady (1999) believed that the suitable temperature range for nitrification was 25ï¡đ35℃, and the nitrification inhibits below 5 ℃ or above 50 ℃. It showed that the temperature requirements of nitrifying microorganisms in wetland soils were different in different temperature belts. The suitable temperature range was the performance of the long-term adaptability of nitrifying microorganisms. Meanwhile, several studies revealed that denitrification could be carried out in a wide temperature range (5ï¡đ70 ℃), and it was positively related to temperature (Fan., 1995). However, the process will be inhibited when the temperature was too high or too low. The average ST in wet season was 27.4°C, conducive to the growth of denitrifying microorganisms, while that in dry season was 8.97°C, and the microbial activity was generally low (Sun et al., 2011). Furthermore, ST fluctuations were low both in wet season and dry season. Therefore, the effect of ST on N2O emissions was masked by other factors, such as moisture content". "Soil carbon source has an important influence on microbial activity. Nitrifying or denitrifying microorganisms need organic matter to provide carbon source during the assimilation of NH3 or NO3-. The high content of organic matter in the soil can promote the abundance of heterotrophic nitrifying bacteria increases, consume dissolved oxygen in the medium, and cause the soil to become more anaerobic, slowing down autotrophic growth nitrifying bacteria. This reduces the nitrification rate, ultimately promoting

N2O release. Enwall et al. (2005) studied the effect of long-term fertilization on soil denitrification microbial action intensity. They found that the soil with long-term organic fertilizer application has a significant increase in organic matter content, and consequently, a significant increase in denitrification activity". "Moreover, incomplete denitrification leads to the accumulation of NO2-N, which is conducive to the N2O release. Meanwhile, due to the weak competitive ability of Nos to electrons, low C:N inhibits the synthesis of Nos, which is also a reason for N2O release". i)table 3: please add the number of samples (n). Reply: We have added the number of samples in table 5. j)line 464 and line 472: I would like to see the long term balance of CO2. Do we have a source or a sink in degraded wetlands considering a longer time span (several years)? Reply: Just like Question g, we cannot calculate the CO2 balance. However, according to the variation trend along the transects and in the longitudinal direction, the wetlands will gradually change into grasslands under the long-term degradation, and are carbon sinks. Meanwhile, theÂăgrasslands have aÂălowerÂăcarbonÂăfixationÂăcapacityÂăthanÂătheÂăwetlands, causing soil carbon loss.

Please also note the supplement to this comment:
https://bg.copernicus.org/preprints/bg-2020-184/bg-2020-184-AC1-supplement.pdf
* * *
**BGD**

[Figure]

Wet season

[Figure]

Dry season

[revised manuscript text omitted]

---

## Author Response (AR1)

March 29, 2021

Dear Editors and Reviewers:

I very much appreciate your efforts and time in reviewing our manuscript.

According to your precious advice and suggestions, we have revised this manuscript thoroughly.

Response to each question from editors and reviewers were listed below.

Thank you very much for your precious time and tremendous efforts in reviewing and supporting this manuscript.

Best Regards,

Xinyu Liu

Inner Mongolia Key Lab of River and lake ecology & Ministry of Education Key Laboratory of Ecology and Resource Use of the Mongolian Plateau

School of Ecology and Environment

Inner Mongolia University

Room 106, Biology Building

No. 235, West University Road, Saihan District, Hohhot

Inner Mongolia 010021, P. R. China

Mobile: +86-13245131615

E-mail: 21815009@mail.imu.edu.cn

Reviewers' comments:

Reviewer #1:

a) Table 1: please provide the number of the samples (n). Moreover, the grain size distributions (or the % age of sand, silt, clay) should be added. Additionally, the saturated volumetric water content and the residual volumetric water content of the soil should be determined.

Reply: We have added the number of the samples (n), annual soil volumetric moisture content for the 0–10 cm and 10–20 soil depth in wet season and in dry season, and the saturated soil moisture content (SSM) in Table 1 and the grain size distributions added in Table 2. However, we don't add the residual volumetric water content, because we cannot measure the matrix suction and draw the pF curve. Residual volumetric water usually obtained by fitting the pF curve with the van genuchten formula. This is another research direction, and we do not have enough theory to study it.

Table 1. Physical and chemical properties (Mean±SD) of soils at various sites within each transect

| Transect | Zone | Samples number | SMC10-V | SMC20-V | Soil C:N | TOC (g·kg⁻¹) | BIO (g) | $\rho_b$ | pH | EC (μs/cm) | SSM (%) |
|---|---|---|---|---|---|---|---|---|---|---|---|
| T1 | Riparian | 12 | 12.16 ± 7.55 | 12.88 ± 12.05 | 12.46 ± 0.91 | 30.16 ± 6.54 | 14.67 ± 5.44 | 1.28 ± 0.07 | 7.25 ± 0.62 | 154.71 ± 23.70 | 47.77 ± 7.04 |
| | Hillslope | 6 | 2.72 ± 0.91 | 5.05 ± 3.09 | 11.41 ± 0.09 | 10.77 ± 4.72 | 6.70 ± 1.48 | 1.45 ± 0.03 | 7.22 ± 0.40 | 82.02 ± 16.37 | 31.02 ± 1.32 |
| T2 | Riparian | 12 | 26.75 ± 19.52 | 12.19 ± 7.82 | 11.70 ± 1.14 | 19.96 ± 5.71 | 24.76 ± 9.65 | 1.23 ± 0.05 | 8.95 ± 0.45 | 303.88 ± 102.16 | 51.21 ± 6.49 |
| | Hillslope | 9 | 5.85 ± 4.82 | 3.03 ± 1.43 | 9.77 ± 0.88 | 14.87 ± 11.21 | 6.10 ± 3.19 | 1.38 ± 0.13 | 8.10 ± 0.55 | 162.97 ± 128.18 | 35.09 ± 6.75 |
| T3 | Riparian | 12 | 28.04 ± 22.95 | 14.53 ± 8.98 | 15.80 ± 4.16 | 22.40 ± 9.69 | 6.37 ± 2.95 | 1.35 ± 0.19 | 9.50 ± 0.67 | 1233.20 ± 829.83 | 47.56 ± 11.65 |
| | L3 | 3 | 116.37 ± 56.91 | 113.36 ± 23.17 | 16.8 ± 0.58 | 36.1 ± 1.84 | 107.75 ± 16.94 | 0.592 ± 0.02 | 8.5 ± 0.17 | 403 ± 57.21 | >100 |
| T4 | Riparian | 12 | 5.42 ± 3.34 | 4.07 ± 4.31 | 12.52 ± 2.06 | 9.96 ± 1.25 | 11.97 ± 4.50 | 1.30 ± 0.08 | 8.84 ± 0.22 | 461.72 ± 314.27 | 44.08 ± 7.07 |

| Transect | Zone | n | SMC10-V | SMC20-V | Soil C:N | TOC | BIO | $\rho_b$ | pH | EC | SSM |
|---|---|---|---|---|---|---|---|---|---|---|---|
| | Hillslope | 6 | 3.35 ± 2.06 | 4.27 ± 1.94 | 9.97 ± 0.50 | 9.65 ± 1.05 | 7.84 ± 2.48 | 1.30 ± 0.09 | 8.23 ± 0.14 | 118.5 ± 8.25 | 39.43 ± 5.55 |
| T5 | Dry lake bed | 12 | 17.47 ± 15.08 | 14.49 ± 13.28 | 63.74 ± 12.93 | 31.41 ± 6.55 | 5.48 ± 2.35 | 1.16 ± 0.10 | 9.88 ± 0.18 | 7320.87 ± 4300.03 | 58.47 ± 7.16 |
| | Lake shore | 9 | 2.64 ± 1.48 | 2.82 ± 1.27 | 15.92 ± 4.71 | 6.35 ± 1.16 | 0 | 1.33 ± 0.09 | 9.41 ± 0.7 | 281.82 ± 162.73 | 37.52 ± 5.34 |

Note: SMC10-V - soil volumetric moisture content in 0-10 cm; SMC20-V - soil volumetric moisture content in 10-20 cm; Soil C:N - soil carbon-nitrogen ratio; TOC - total soil organic carbon; BIO - aboveground biomass; $\rho_b$ - soil bulk density; pH - soil pH; EC - soil electrical conductivity; SSM - saturated soil moisture.

Table 2. soil particle composition of soils at various sites within each transect

| Transect | Zone | soil particle composition | | |
|---|---|---|---|---|
| | | Clay % (<0.002 mm) | Silt % (0.02~0.002 mm) | Sand (2.0 ~0.02 mm) |
| T1 | Riparian | 2.5 | 2.7 | 94.8 |
| | Hillslope | 9.6 | 6.1 | 85.3 |
| T2 | Riparian | 5.5 | 5.8 | 90.7 |
| | Hillslope | 10.8 | 8.6 | 80.6 |
| T3 | Riparian | 4.1 | 1.1 | 94.8 |
| T4 | Riparian | 11.4 | 1.5 | 87.1 |
| | Hillslope | 12.7 | 5.9 | 81.4 |
| T5 | Lake shore | 5.1 | 2.1 | 92.8 |
| | Dry lake bed | 46.1 | 4.8 | 49.1 |

b) Fig. 3: It is not clear if the SMC(%) is based on volume or mass. Also in the text the numbers for SMC are not clear. I suppose, the values are gravitational SMCs. It is important that SMC is related to the soil water capacity and the pF curve of the soils. Therefore, relative saturation would be a better measure. Alternatively, the authors can define the field capacity and the saturation values of the different soils.

Reply: SMC stands for soil mass moisture content, which has been indicated on line 144. We have

rewritten the contents of the SMC, marking SMC10 and SMC20 as following:

"The temporal and spatial variations in SMC10 in the following order: wet season > dry season and riparian wetlands > hillslope grasslands (Fig. 3a, c, e). Similar variations were observed in SMC20 (Fig. 3b, d, f). The average SMC10 and SMC20 in the continuous river transects in the riparian zones (37.44% in wet season and 19.40% in dry season; 25.96% in wet season and 17.39% in dry season) were higher than those in the hillslope grasslands (9.12% in wet season and 4.15% in dry season; 6.51% in wet season and 5.96% in dry season). During the study period, both SMC10 and SMC20 changed as the distance from the river increased, and the highest value was observed at the near-stream sites (L1 and R1). SMC10 fluctuations were low in the intermittent transect compared to the upstream transects, with a mean value of 11.79% in wet season and 3.72% in dry season in the riparian areas. The mean SMC10 in the hillslopes was 6.58% in wet season and 2.86% in dry season. SMC20 showed similar fluctuation, 7.22% in wet season and 2.98% in dry season in the riparian areas and 7.56% in wet season and 4.4% in dry season in the hillslopes. In transect T5, average SMC10 and SMC20 at the center of the lake (29.00% in wet season and 13.36% in dry season; 29.30% in wet season and 9.69% in dry season) were higher than those along the lake shore (4.90% in wet season and 3.13% in dry season; 3.34% in wet season and 5.22% in dry season)".

c) Fig. 4) please integrate into the figures an improved legend. Then you can skip the lenghty text of fig.4.

Reply: We have revised the legend in fig.4 and shortened the lengthy text of fig.4.

Wet season

[Figure]

Dry season

[Figure]

Fig. 4 Soil temperature (ST) at soil depths of 0–10 cm (ST10) and 10–20 cm (ST20) for transects T1–T5 in wet season and dry season. Error bars represent the SD about the mean.

d) Fig. 6) please indicate Riparian wetlands and hillslope grasslands directly in the figures. Then you can shorten the lengthy text of fig. 6.

Reply: We have indicated "Riparian wetlands" and "Hillslope grasslands" in fig.6 and shortened the lengthy text of fig.6.

[Figure]

[Figure]

Fig. 6 Spatiotemporal patterns of $CO_2$ (first line), $CH_4$ (second line), and $N_2O$ (third line) emissions ($F$) in the upstream (T1, T2, and T3) and downstream areas (T4 and T5). Bars are the mean values for each transect and error bars are the standard errors.

e) line 292 and line 300/ line 301: SMC values of 40 to 60%... This must be related to the soil, because SMC is a function of suction (matrix potential).

Reply: Yes, this is a very complex subject, and the soil's permeability is difficult to determine. This is another research direction, and we do not have enough theory to study it. So, we determined soil mass moisture content simply using experimental methods to illustrate the relationship between SMC and GHGs emissions.

f) line 312: What means: "SMC was above the saturated water content"? This is not possible.

Reply: Sorry for confusing you. "SMC was above the saturated water content" means that the soil reaches saturation. Thus, we have revised the sentence to "When SMC increases to the saturated water content or is in a flooded state, the system was an anaerobic environment", which has been indicated on line 355.

g) Chapter 4.1.3: It would be beneficial for the understanding, if the authors can calculate $CO_2$ balances. Is the balance of photosynthesis and respiration / emission positive or negative?

Reply: The paper uses the static dark chamber method to measure the ecosystem's respiration and

discusses the "emission" part of greenhouse gases. The "absorption" is not measured, so the $CO_2$ balance cannot be calculated. This is a very good suggestion that can be studied in the future. Generally, photosynthesis in healthy wetlands is more significant than respiration, conducive to the accumulation of organic matter. During the wetlands' degradation, the plant community and microbial composition change, the biomass is reduced, and photosynthesis is minor than respiration, causing carbon loss in the wetlands. After wetlands completely degraded, photosynthesis is more excellent than respiration, reaching a new balance. However, compared with a healthy wetland, the accumulation of organic matter is significantly reduced.

h) The nitrification / denitrification description is too vague. Please insert the formulas of the nitrification / denitrification processes and determine its relation / quantification.

Reply: We have added the formula and modified it in various parts of 4.1.1, 4.1.2, and 4.1.3.

[revised manuscript text omitted]

j) line 464 and line 472: I would like to see the long term balance of $CO_2$. Do we have a source or a sink in degraded wetlands considering a longer time span (several years)?

Reply: Just like Question g, we cannot calculate the $CO_2$ balance. However, according to the variation trend along the transects and in the longitudinal direction, the wetlands will gradually change into grasslands under the long-term degradation, and are carbon sinks. Meanwhile, the grasslands have a lower carbon fixation capacity than the wetlands, causing soil carbon loss.

Reviewers' comments:

Reviewer #2:

**Title** Consider removing the word "river" as riparian wetlands already define wetlands on stream and riverbanks.

Reply: We have removed the word "river" from the title, and the title has been changed to "Greenhouse gases emissions from riparian wetlands: An example from the Inner Mongolia grassland region in China".

**Abstract**

Ln 13-15: Consider reversing the sentence to give details on the direct link between riparian wetlands and climate change.

Reply: We have reversed the order of two sentences.

"Riparian wetland drying/degradation is increasingly sensitive to global warming and human activities, and contributes to climate change. Riparian wetlands play a significant role in regulating carbon and nitrogen cycles".

**Introduction**

Ln 56: Remove the word "the" in "the nature: : :.."

Reply: We have removed the word "the" in "the nature".

"Wetlands are increasingly recognized as an essential part of nature,..."

Ln 79: Remove the word "the" in "at the local: : :.."

Reply: We have removed the word "the" in "at the local".

"Moreover, it is necessary to estimate the changes in GHG emissions as a result of wetland degradation at local and global scales."

**Materials and methods**

Ln 117: Replace "for" with "from"

Reply: We have replaced the word "for" with "from".

"Each sampling point from T1-T5 was extended from the river to both sides, ..."

Ln 137: Consider replacing "the" with "a" in the reservoir bag.

Reply: We have replaced the word "the" with "a".

"The gas samples were stored in a reservoir bag"

Ln 139: : : :.and or or for the sampling times. If "and", were they averaged for the day? It is a bit not clear now.

Reply: The sampling time is 9:00-11:00 a.m. or 3:00-5:00 p.m. when we conducted the measurement in the different sampling sites of the same transect.

"The measurements were scheduled for 9:00-11:00 a.m. or 3:00-5:00 p.m."

Ln 141: : : :.oven-dried. .

Reply: We have corrected the spelling of the word.

"oven-dried in the laboratory to obtain aboveground biomass"

Ln 148: Figure 2, colors of the site labels are too difficult to see, consider using more contrasting colors.

Reply: We have modified the colors of the site labels in Figure 2.

[Figure]

Fig. 2 Distributions of sampling points in transects T1–T5 (The images are authors' own)

Ln 155: Indicate whether they are means and SD or SE in table caption.

Reply: The numbers in Table 1 are Mean±SD, and SD has been labbed in the table caption.

Table 1. Physical and chemical properties (Mean±SD) of soils at various sites within each transect

| Transect | Zone | Samples number | SMC10-V | SMC20-V | Soil C:N | TOC (g·kg⁻¹) | BIO (g) | $\rho_b$ | pH | EC (μs/cm) | SSM (%) |
|---|---|---|---|---|---|---|---|---|---|---|---|
| T1 | Riparian | 12 | 12.16 ± 7.55 | 12.88 ± 12.05 | 12.46 ± 0.91 | 30.16 ± 6.54 | 14.67 ± 5.44 | 1.28 ± 0.07 | 7.25 ± 0.62 | 154.71 ± 23.70 | 47.77 ± 7.04 |
| | Hillslope | 6 | 2.72 ± 0.91 | 5.05 ± 3.09 | 11.41 ± 0.09 | 10.77 ± 4.72 | 6.70 ± 1.48 | 1.45 ± 0.03 | 7.22 ± 0.40 | 82.02 ± 16.37 | 31.02 ± 1.32 |
| T2 | Riparian | 12 | 26.75 ± 19.52 | 12.19 ± 7.82 | 11.70 ± 1.14 | 19.96 ± 5.71 | 24.76 ± 9.65 | 1.23 ± 0.05 | 8.95 ± 0.45 | 303.88 ± 102.16 | 51.21 ± 6.49 |
| | Hillslope | 9 | 5.85 ± 4.82 | 3.03 ± 1.43 | 9.77 ± 0.88 | 14.87 ± 11.21 | 6.10 ± 3.19 | 1.38 ± 0.13 | 8.10 ± 0.55 | 162.97 ± 128.18 | 35.09 ± 6.75 |
| T3 | Riparian | 12 | 28.04 ± 22.95 | 14.53 ± 8.98 | 15.80 ± 4.16 | 22.40 ± 9.69 | 6.37 ± 2.95 | 1.35± 0.19 | 9.50 ± 0.67 | 1233.20 ± 829.83 | 47.56 ± 11.65 |
| | L3 | 3 | 116.37 ± 56.91 | 113.36 ± 23.17 | 16.8 ± 0.58 | 36.1 ± 1.84 | 107.75 ±16.94 | 0.592 ± 0.02 | 8.5 ± 0.17 | 403 ± 57.21 | >100 |
| T4 | Riparian | 12 | 5.42 ± 3.34 | 4.07 ± 4.31 | 12.52 ± 2.06 | 9.96 ± 1.25 | 11.97 ± 4.50 | 1.30 ± 0.08 | 8.84 ± 0.22 | 461.72 ± 314.27 | 44.08 ± 7.07 |
| | Hillslope | 6 | 3.35 ± 2.06 | 4.27 ± 1.94 | 9.97 ± 0.50 | 9.65 ± 1.05 | 7.84 ± 2.48 | 1.30 ± 0.09 | 8.23 ± 0.14 | 118.5 ± 8.25 | 39.43 ± 5.55 |
| T5 | Dry lake bed | 12 | 17.47 ± 15.08 | 14.49 ± 13.28 | 63.74 ± 12.93 | 31.41 ± 6.55 | 5.48 ± 2.35 | 1.16 ± 0.10 | 9.88 ± 0.18 | 7320.87 ± 4300.03 | 58.47 ± 7.16 |
| | Lake shore | 9 | 2.64 ± 1.48 | 2.82 ± 1.27 | 15.92 ± 4.71 | 6.35 ± 1.16 | 0 | 1.33 ± 0.09 | 9.41 ± 0.7 | 281.82 ± 162.73 | 37.52 ± 5.34 |

Note: SMC10-V - soil volumetric moisture content in 0-10 cm; SMC20-V - soil volumetric moisture content in 10-20 cm; Soil C:N - soil carbon-nitrogen ratio; TOC - total soil organic carbon; BIO - aboveground biomass; $\rho_b$ - soil bulk density; pH - soil pH; EC - soil electrical conductivity; SSM - saturated soil moisture.

Ln 166: Missing section on what statistical tests were used for the analysis of the results.

Reply: We have added the missing section of statistical analysis in line 169 as following:

"2.4 Statistical Analysis

All statistical analyses were performed using SPSS for Windows version 18.0 (SPSS Inc., Chicago, IL, USA). Statistical significance was set at $P<0.05$. Pearson correlation analysis was conducted to estimate the relationships between GHGs fluxes and environmental variables. A Wilcoxon test was used to determine the difference of GHGs fluxes in two seasons."

**Results**

Ln 169: Variations in SMC?

Reply: We have revised this sentence as follows:

"The temporal and spatial variations in SMC10 in the following order: wet season (August) > dry season (October), and riparian wetlands > hillslope grasslands (Fig. 3a, c, e)".

Ln 169- 171: Confusing, consider revising the sentence to make it clearer.

Reply: We have rewritten this sentence as following:

"The temporal and spatial variations in SMC10 in the following order: wet season > dry season, and riparian wetlands > hillslope grasslands (Fig. 3a, c, e). Similar variations were observed in SMC20 (Fig. 3b, d, f)".

Ln 173: Cite the section in figure 3 to enable the reader follow easily the results section.

Reply: We have rewritten this sentence in line 189-line 204.

"The temporal and spatial variations in SMC10 in the following order: wet season > dry season and riparian wetlands > hillslope grasslands (Fig. 3a, c, e). Similar variations were observed in SMC20 (Fig. 3b, d, f). The average SMC10 and SMC20 in the continuous river transects in the riparian zones (37.44% in wet season and 19.40% in dry season; 25.96% in wet season and 17.39% in dry season) were higher than those in the hillslope grasslands (9.12% in wet season and 4.15% in dry season; 6.51% in wet season and 5.96% in dry season). During the study period, both SMC10 and SMC20 changed as the distance from the river increased, and the highest value was observed at

the near-stream sites (L1 and R1). SMC10 fluctuations were low in the intermittent transect compared to the upstream transects, with a mean value of 11.79% in wet season and 3.72% in dry season in the riparian areas. The mean SMC10 in the hillslopes was 6.58% in wet season and 2.86% in dry season. SMC20 showed similar fluctuation, 7.22% in wet season and 2.98% in dry season in the riparian areas and 7.56% in wet season and 4.4% in dry season in the hillslopes. In transect T5, average SMC10 and SMC20 at the center of the lake (29.00% in wet season and 13.36% in dry season; 29.30% in wet season and 9.69% in dry season) were higher than those along the lake shore (4.90% in wet season and 3.13% in dry season; 3.34% in wet season and 5.22% in dry season)".

Ln 180: Consider indicating on the graphs season information to make it less confusing. i.e add wet on top of the first two graphs and dry on the second pair of graphs. What are the error bars? Standard errors? Also throughout all the manuscript, consider using wet and dry instead of the months as it gives a more direct link to the hydrological conditions of the riparian wetlands.

Reply: We have reworked Figure 3 according to your suggestion. The error bars are standard deviations which was explained in the title of Figure 3. Meanwhile, we have replaced the "August" with "wet season" and replaced the word "October" with "dry season".

Wet season

[Figure]

Dry season

[Figure]

Fig. 3 Soil mass moisture contents (SMCs) at soil depths of 0–10 cm (SMC10) and 10–20 cm (SMC20) for transects T1–T5 in wet season and dry season. Error bars represent the SD about the mean.

Ln 193: Not clearly seen in the graphs, maybe change the shapes of the points within the riparian region.

Reply: We have reworked Figure 4.

Wet season

[Figure]

Dry season

[Figure]

Fig. 4 Soil temperature (ST) at soil depths of 0–10 cm (ST10) and 10–20 cm (ST20) for transects T1–T5 in wet season and dry season. Error bars represent the SD about the mean.

Ln 196: Same comments as SMC on the visuals.

Reply: We have reworked Figure 3.

Wet season

[Figure]

[Figure]

Fig. 3 Soil mass moisture contents (SMCs) at soil depths of 0–10 cm (SMC10) and 10–20 cm (SMC20) for transects T1–T5 in wet season and dry season. Error bars represent the SD about the mean.

LN 226: What statistical tests were used to show differences in the two seasons? This information is missing in the figure and in the text.

Reply: We used the Wilcoxon test to determine the difference of GHGs fluxes in the two seasons and showed the results in Table 3.

"2.4 Statistical Analysis

All statistical analyses were performed using SPSS for Windows version 18.0 (SPSS Inc., Chicago, IL, USA). Statistical significance was set at $P < 0.05$. Pearson correlation analysis was conducted to estimate the relationships between GHGs fluxes and environmental variables. **A Wilcoxon test was used to determine the difference of GHGs fluxes in two seasons.**"

Ln 247: Figure is stretched vertically. Check this for all figures to ensure the aspect ratio is maintained

when adding them in the document.

[Figure]

Fig. 6 Spatiotemporal patterns of $CO_2$ (first line), $CH_4$ (second line), and $N_2O$ (third line) emissions ($F$) in the upstream (T1, T2, and T3) and downstream areas (T4 and T5). Bars are the mean values for each transect and error bars are the standard errors.

**Discussion**

Ln 282: The discussion includes results not shown in the results section. Consider shifting some of the results in the discussion to the results part of the manuscript.

Ln 288: Indicate whether the correlation is positive or negative.

Reply: The correlation between SMC and GHGs is positive.

"Table 4, SMC10 is positive correlated with $CO_2$ emissions (P < 0.05), SMC10 and SMC20 are significantly positive correlated with $CH_4$ emissions (P < 0.01), and SMC10 and SMC20 are highly positive correlated with $N_2O$ emissions (P < 0.05 and P < 0.01, respectively)".

Ln 292: Give more details on the mechanism that links SMC to $CO_2$ fluxes that the authors found, and how it links with your findings.

Reply: We added the mechanism of SMC on $CO_2$ emissions.

"Typically, the optimal SMC values associated with $CO_2$ emissions in riparian wetlands range from 40 to 60% (Sjögersten et al., 2006), creating better soil aeration and improving soil microorganisms' activity and the respiration of plant roots, thereby promoting $CO_2$ emissions, whereas excessive SMC reduces soil gas transfer due to the formation of an anaerobic environment in the soil, and microbial activity is lower, favoring the accumulation of organic matter (Hui., 2014). On the contrary, the SMC of hillslope grasslands is less than 10%. Low soil moisture inhibits the growth of vegetation with few vegetation residues and litters. Meanwhile, low soil moisture is not conducive to the survival of soil microorganisms, leading to a decrease in $CO_2$ emissions than to those in riparian zones (Moldrup et al., 2000; Hui., 2014)".

Ln 296: How was this shown in the results? Seems rather speculative. Possibly give ranges based on other studies and link them with your study as shown in Table 1.

Reply: We have re-written this sentence as following:

"The changes in $CO_2$ emissions in transect T5 were contrary to the change in the SMC10 and SMC20 likely because the optimal range of soil C:N is between 10-12 (Pierzynski et al., 1994), but the value in the dry lake bed of T5 is higher than 60, high soil C:N resulted in nitrogen limitation in the process of decomposition of organic matter by microorganisms. Furtherly, other sediment properties (like Soil pH>9.5) for this transect were not conducive to the survival of microorganisms (Table 1), and the increase in SMC did not increase the respiration activity of microorganisms".

Ln 308: You mean aerobic decomposition.

Reply: Yes, aerobic decomposition. As the SMC decreases, the soil oxide layer expands, and $CH_4$

emissions change from source to sink.

Ln 311: Is this shown in the results section? Not clear what value of SMC indicates the saturation water content.

Reply: We have added the soil's saturated water content to Table 1 in the result part, and linked it to the discussion part in line 351.

"Generally, when SMC was below the saturated water content, the microorganisms were in an aerobic environment, and $N_2O$ mainly came from the nitrification reaction. $N_2O$ emissions increases with the increase of SMC (Niu et al., 2017; Yu et al., 2006). In our study, the sampling sites with higher SMC (riparian zones and some hillslope grassland zones in the upstream transects) have higher $N_2O$ emissions. When SMC increases to the saturated water content or is in a flooded state, the system was an anaerobic environment, and the Nos activity was higher due to excessively high SMC, which was conducive to denitrification and eventually produced $N_2$ (Niu et al., 2017; Yu et al., 2006), such as site L1 in transect T3 in this study. Ulrike et al. (2004) showed that denitrification was the main process under flooded soil conditions in wetland soils, and the release of $N_2$ exceeds $N_2O$".

Ln 313: More details on how the Niu et al 2017 study relates to your study.

Reply: We have added this part's content as following:

"Generally, when SMC was below the saturated water content, the microorganisms were in an aerobic environment, and $N_2O$ mainly came from the nitrification reaction. $N_2O$ emissions increases with the increase of SMC (Niu et al., 2017; Yu et al., 2006)", "When SMC increases to the saturated water content or is in a flooded state, the system was an anaerobic environment, and the Nos activity was higher due to excessively high SMC, which was conducive to denitrification and eventually produced $N_2$ (Niu et al., 2017; Yu et al., 2006)".

Ln 316: What mechanism links increased SMC to higher $N_2O$ fluxes? Currently the information is missing.

Reply: We have added the content of this part and showed in line 351-361.

"Generally, when SMC was below the saturated water content, the microorganisms were in an aerobic environment, and $N_2O$ mainly came from the nitrification reaction. $N_2O$ emissions increases with the increase of SMC (Niu et al., 2017; Yu et al., 2006). In our study, the sampling sites with higher SMC (riparian zones and some hillslope grassland zones in the upstream transects) have higher $N_2O$ emissions. When SMC increases to the saturated water content or is in a flooded state, the system was an anaerobic environment, and the Nos activity was higher due to excessively high SMC, which was conducive to denitrification and eventually produced $N_2$ (Niu et al., 2017; Yu et al., 2006), such as site L1 in transect T3 in this study. Ulrike et al. (2004) showed that denitrification was the main process under flooded soil conditions in wetland soils, and the release of $N_2$ exceeds $N_2O$. These findings are consistent with those of Liu et al. (2003), who showed that SMC is an essential factor affecting $N_2O$ emissions".

Nitrification:

$$NH_4^+ \xrightarrow{AMO} NH_2OH \longrightarrow [NOH] \xrightarrow{HAO} NO_2^- \xrightarrow{NXR} NO_3^-$$

$$\underset{Nir}{} \qquad \downarrow \qquad \overset{Nor}{\nwarrow} \qquad \downarrow$$

$$N_2O \longleftarrow NO \tag{4}$$

Denitrification:

$$NO_3^- \xrightarrow{Nar} NO_2^- \xrightarrow{Nir} NO \xrightarrow{Nor} N_2O \xrightarrow{Nos} N_2 \tag{5}$$

The enzymes involved in the formula include Ammonia monooxygenase (AMO), Hydroxylamine oxidase (HAO), Nitrite REDOX enzyme (HAO), nitrate reductase (Nar), nitrite reductase (Nir), Nitric oxide reductase (Nor) and Nitrous oxide reductase (Nos).

Ln 330: Confusing as you say its important at the start of the paragraph.

Reply: Sorry, there are indeed problems in our consideration, and we have deleted the contradictions. Temperature is an important factor that affects $CH_4$ emissions. However, temperature was not significantly corelated to $CH_4$ emissions in our study, likely because SMC could be more critical than temperature in our study region with very dry climate.

Ln 336: Consider replacing the growing season to either August or October. Currently it is not clear

which season is the growing season for a reader not familiar with the region of study.

Reply: We have replaced the word "growing season" with "August".

"However, the wetlands maintained a state without water accumulation on the soil surface in August, which was conducive to the oxidative absorption of $CH_4$. SMC thus masked the effect of ST on $CH_4$ emissions".

Ln 364: Do soil nutrients mean SOC. Not clear at the moment.

Reply: Soil nutrients refer to the nutrient elements necessary for plant growth provided by the soil. However, in this study, we do not measure various mineral elements' content, e.g., K, Ca, Mg, Fe, P, etc. So, soil nutrients are simply defined as C and N beneficial to plants and microorganisms' growth and denoted by TOC and soil C: N in Table 1.

Ln 380: remove "the" in "the soil C:N: : :.."

Reply: We have removed the word "the" in "the soil C:N".

"Soil C:N ratio refers to the ratio of biodegradable carbonaceous organic matter and nitrogenous matter in the soil".

Ln 381: TOC is also part of the C:N ratio. Elaborate more on the disentanglement between the two in the point you are making.

Reply: We have added this part's content as following:

"TOC decomposition provides energy for microbial activity, while the C:N ratio affects the decomposition of organic matter by soil microorganisms (Gholz et al., 2010). TOC has a weak positive correlation with $CO_2$ emissions (P>0.05), but soil C:N has a significant negative correlation with $CO_2$ emissions (P<0.05), indicating that nitrogen has a limiting effect on soil respiration by affecting microbial metabolism. Liu et al. (2019) reported that N addition promoted $CO_2$ emissions from wetlands soil, and the effect of organic N input was significantly higher than those of inorganic N input. Organic carbon provides a carbon source for the growth of plants and microorganisms, which boosts their respiration".

Ln 384: But the statistics show the correlation with TOC is not significant.

Reply: We have added this part's content as following:

"TOC has a weak positive correlation with $CO_2$ emissions (P>0.05), but soil C:N has a significant negative correlation with $CO_2$ emissions (P<0.05), indicating that nitrogen has a limiting effect on soil respiration by affecting microbial metabolism. Liu et al. (2019) reported that N addition promoted $CO_2$ emissions from wetlands soil, and the effect of organic N input was significantly higher than those of inorganic N input. Organic carbon provides a carbon source for the growth of plants and microorganisms, which boosts their respiration".

Ln 389: Elaborate more how this promotes $N_2O$ release.

Reply: We have added a more detailed explanation about how organic carbon promoting $N_2O$ emissions.

"Most heterotrophic microorganisms use soil organic matter as carbon and electron donors (Morley N and Baggs E M., 2010). Soil carbon source has an important influence on microbial activity. Nitrifying or denitrifying microorganisms need organic matter to provide carbon source during the assimilation of $NH_3$ or $NO^{3-}$. The high content of organic matter in the soil can promote the abundance of heterotrophic nitrifying bacteria increases, consume dissolved oxygen in the medium, and cause the soil to become more anaerobic, slowing down autotrophic growth nitrifying bacteria. This reduces the nitrification rate, ultimately promoting $N_2O$ release. Enwall et al. (2005) studied the effect of long-term fertilization on soil denitrification microbial action intensity. They found that the soil with long-term organic fertilizer application has a significant increase in organic matter content, and consequently, a significant increase in denitrification activity".

Ln 403: More description required for the table. For example if the values given are correlation coefficients and the type of correlation test used.

Reply: We have added the missing content to "notes" under Table 4, "The analysis method used in the table is Pearson correlation analysis, and the numbers represent Pearson correlation coefficients".

Table 4. Correlations between $CO_2$, $CH_4$, and $N_2O$ emissions and impact factors ($n = 62$)

| GHG flux | ST10 | ST20 | SMC10 | SMC20 | TOC | $\rho_b$ | C:N | pH | EC | BIO |
|----------|------|------|-------|-------|-----|----------|-----|----|----|-----|

| | | | | | | | | | | |
|---|---|---|---|---|---|---|---|---|---|---|
| $CO_2$ | 0.634** | 0.592** | 0.307* | 0.216 | 0.393 | −0.463** | −0.289* | −0.350** | −0.251* | 0.491* |
| $CH_4$ | −0.029 | −0.051 | 0.346** | 0.353** | −0.02 | −0.129 | −0.156 | −0.127 | −0.107 | 0.607** |
| $N_2O$ | 0.127 | 0.118 | 0.304* | 0.356** | 0.493* | −0.194 | 0.311* | 0.137 | 0.504** | 0.251 |

Note: 1. The analysis method used in the table is Pearson correlation analysis, and the numbers represent Pearson correlation coefficients.

Ln 417: Table 1 also shows higher C:N ratios in riparian soils.

Reply: Table 1 shows that high C:N ratios only occurs in the dry lake bed of transect T5, but The site range mentioned in this study is "the upstream riparian wetlands",in other words, is T1, T2, T3 transect riparian wetlands.

Ln 422: Elaborate more on the link between $CO_2$ concentrations and nitrification denitrification processes to make it clearer for the reader.

Reply: We have added an explanation about $CO_2$ concentration and nitrification-denitrification processes in lines 491-496.

"The $N_2O$ emissions showed spatial patterns similar to those of the $CO_2$ emissions because the $CO_2$ concentrations were closely related to nitrification and denitrification processes. High $CO_2$ concentrations can promote the carbon and nitrogen cycles in soil (Azam et al., 2005), increasing below ground C allocation associated with increased root biomass, root turnover, and root exudation in elevated $pCO_2$ plants provided the energy for denitrification in the presence of high available N, or that there was increased $O_2$ consumption under elevated $\rho CO_2$ (Baggs et al., 2003)".

Ln 432: use "and" instead of "but" as the latter indicates differences in the findings of the two studies. Is that the case? If yes, consider reversing the sentence to clearly bring it out.

Reply: We have replaced the word "but" with "and".

"Jacinthe et al. (2015) reported that inundated grassland-dominated riparian wetlands were $CH_4$ sinks (-1.08±0.22 kg·$CH_4$-C ha$^{-1}$·yr$^{-1}$), and Lu et al. (2015) also indicated that grasslands were $CH_4$ sinks".

Ln 442: remove "the" in as the sources of: : :.

Reply: We have removed the word "the" in "as the sources of".

"Moreover, the upper riparian wetlands acted as source of $CH_4$ emissions".

Ln 466: Was the soil carbon in the degraded wetlands lost through aerobic decomposition. Give more details on the mechanism.

Reply: We have added the explanation about wetland degradation caused the loss of the soil carbon pool and weakened the wetland carbon source/sink function in lines 551-567.

"The wetland degradation first resulted in the continuous reduction of SMC, which led to the deepening of the wetland's aerobic layer thickness. Besides, SMC could affect ST's change and thus transformed $CH_4$ emissions from a source to a sink by affecting methanogens' activity (Yan et al., 2018). Secondly, the reduction of SMC impeded aboveground plants' physiological activities and inhabited related enzymes' activities in the respiration process. Meanwhile, various enzyme reactions of underground microorganisms under water stress influence and reduced $CO_2$ emissions (Zhang et al., 2017). Finally, after wetland degradation, long-term drought caused too low SMC, which was not conducive to the growth of nitrifying and denitrifying bacteria, which caused the transformation of $N_2O$ emissions from source to sink (Zhu et al., 2013). Table 1 shows that soil TOC in the upstream transects (average: 25.1 g·kg$^{-1}$) is higher than that in the downstream transects (average: 8.41 g·kg$^{-1}$). The relatively low SMC and the aerobic environment were conducive to the mineralization and decomposition of TOC. The degradation of plants in the wetlands led to the gradual reduction of BIO. Ultimately, the plant carbon source input of the degraded wetlands decreased, and the bare land temperature increased due to the reduced plant shelter. This accelerated the decomposition of TOC, leading to its decrease. This result indicates that wetland degradation caused the soil carbon pool's loss and weakened the wetland carbon source/sink function. These results are in agreement with those of Xia (2017)".

Conclusion

Ln 486: Comparison of the source strengths of the three gases expressed as GWP not presented in the graphs. This may show more clearly that $CO_2$ contributed more than the other two GHG. Consider adding it.

Reply: We have added the cumulative annual emission flux and global warming potential of GHGs in riparian wetlands and grasslands.

"The annual cumulative emissions were calculated using Eq. 2 (Whiting G and Chanton J., 2001)

$$M = \sum \frac{F_{i+1} + F_I}{2} \times (t_{i+1} - t_i) \times 24$$

(2)

Where M denotes the total cumulative emissions of $CO_2$, $CH_4$, or $N_2O$ (kg·hm$^2$), $F$ is the emission flux of $CO_2$, $CH_4$, or $N_2O$, i is the sampling frequency, $t_{i+1}$-$t_i$ represents the interval between two adjacent measurement dates.

In this study, a 100-year scale was selected to calculate the global warming potential (GWP) of soil $CH_4$ and $N_2O$ emissions (Whiting G and Chanton J., 2001):

$$GWP = 1 \times [CO_2] + 25 \times [CH_4] + 298 \times [N_2O]$$

(3)

Where 25 and 298 are GWP multiples of $CH_4$ and $N_2O$ relative to $CO_2$ on a 100-year time scale, respectively".

"We roughly estimated the annual cumulative emissions of $CO_2$, $CH_4$, and $N_2O$ from riparian wetlands and hillslope grasslands around the Xilin River Basin, and further calculated its global warming potential. Table 6 indicated that annual cumulative emissions of $CO_2$ and $CH_4$ decreased in the following order: upstream riparian wetlands > downstream riparian wetlands > hillslope grasslands, and $N_2O$ in the following order: upstream riparian wetlands > hillslope grasslands > downstream riparian wetlands. In this study, we used the static dark-box method to measure $CO_2$ emissions, which does not consider the absorption and fixation of $CO_2$ by plants' photosynthesis. Therefore, the total annual cumulative $CO_2$ emissions are high. This result clearly showed that $CO_2$ contributed more than $CH_4$ and $N_2O$ to global warming. The GWP depends on the cumulative emissions of the GHGs. GWP is shown as (Table 6): upstream riparian wetlands (13474.91 kg/hm$^2$) > downstream riparian wetlands (8974.12 kg/hm$^2$) > hillslope grasslands (8351.24 kg/hm$^2$). Therefore, both riparian wetlands and grasslands are the "sources" of GHGs on a 100-year time scale. The source strength of wetlands is higher than grasslands, further indicating that riparian wetlands

are the hotspots of GHG emissions".

Table 6 Cumulative annual emission flux and global warming potential of GHGs in riparian wetlands and grasslands

| Sample plot | $CO_2$/kg/hm$^2$ | $CH_4$/kg/hm$^2$ | $N_2O$/kg/hm$^2$ | GWP/$CO_2$ kg hm$^2$ |
|---|---|---|---|---|
| Wetlands of upstream transects (T1, T2, and T3) | 13092.8±5378.16 | 12.36±26.40 | 0.25±0.23 | 13474.91±5828.68 |
| Wetlands of downstream transects (T4 and T5) | 9093.47±4831.82 | -1.68±3.23 | -0.26±0.40 | 8974.12±4912.75 |
| Hillslope grasslands of all transects | 8412.26±1614.26 | -2.55±3.12 | 0.01±0.20 | 8351.24±1648.22 |

---

## Editor Decision (ED1)

[revised manuscript text omitted]

Riveros-Iregui D.A., Mcglynn B.L., Pacific V.J., Epstein H.E., Welsch D.L.: Soil $CO_2$ Efflux

Variability in Complex Terrain: Towards Estimation of Watershed-Level Rates, AGU Fall

Meeting Abstracts, 2007, B21D-04, 2007.

Sjögersten S., Wal R V.D., Woodin S.J.: Small-scale hydrological variation determines landscape

$CO_2$ fluxes in the high Arctic, Biogeochemistry, 80, 205–216, doi 10.2307/20456398, 2006.

Sun Y., Wu H., Wang Y.: The influence factors on $N_2O$ emissions from nirification and denitrification reaction, Ecology and Environmental Sciences, 20, 384–388, doi

10.1631/jzus.B1000275, 2011.

Tong C., Wu J., Yong S., Yang J., Yong W.: A landscape-scale assessment of steppe degradation in the Xilin River Basin, Inner Mongolia, China, Journal of Arid Environments, 59, 133–149, doi

10.1016/j.jaridenv.2004.01.004, 2004.

Ulrike R., Jürgen A., Rolf R., Wolfgang M.: Nitrate removal from drained and reflooded fen soils affected by soil N transformation processes and plant uptake, Soil Biology and Biochemistry, 36,

77-90, doi 10.1016/j.soilbio.2003.08.021, 2004.

Waddington J.M., Roulet N.T.: Carbon balance of a boreal patterned peatland, Global Change

Biology, 6, 87–97, doi 10.1046/j.1365-2486.2000.00283.x, 2000.

Whiting G.J., Chanton J.P.: Greenhouse earbon balance of wetlands: methane emission versus carbon sequestration, Tellus B, 53, 521-528, doi 10.3402/tellusb.v53i5.16628, 2001.

WMO.: WMO Statement on the State of the Global Climate in 2017, World Meteorological

Organization, 2018.

Xi X., Zhu Z., Hao X.: Spatial variability of soil organic carbon in Xilin River Basin, Research of

Soil and Water Conservation, 24, 97–104, 2017.

Xia P., Yu L., Kou Y., Deng H., Liu J.: Distribution characteristics of soil organic carbon and its relationship with enzyme activity in the Caohai wetland of the Guizhou Plateau, Acta Scientiae

Circumstantial, 37, 1479–1485, doi 10.13671/j.hjkxxb.2016.0129, 2017.

Xu H., Cai Z., Yagi K.: Methane Production Potentials of Rice Paddy Soils and Its Affecting

Factors, Acta Pedologica Sinica, 45, 98–104, doi 10.1163/156939308783122788, 2008.

Yan L., Zhang X., Wang J., Li Y., Wu H., Kang X.; Drainage effects on carbon flux and carbon storage in swamps, marshes, and peatlands, Chin J Appl Environ Biol, 24, 1023–1031, doi

10.19675/j.cnki.1006-687x.2017.11031, 2018.

Yu P., Zhang J., Lin C.: Progress of influence factors on $N_2O$ emission in farmland soil,

Environment and sustainable development, 20–22, 2006.

Zhang D.: Effects of Different Grazing Intensities on Greenhouse Gases Flux in Typical Steppe of

Inner Mongolia, Inner Mongolia University, 2019.

Zhang Y., Hao Y., Cui L., Li W., Zhang X., Zhang M., Li L., Yang S., Kang X.: Effects of extreme drought on $CO_2$ fluxes of Zoige alpine peatland, Journal of University of Chinese

Academy of Sciences, 34, 462-470, 2017.

Zhang Z., Hua L., Yin X., Hua L., Gao J.: Nitrous oxide emission from agricultural soil land some influence factors, Journal of Capital Normal University: Natural Science Edition, 26, 114–120,

2005.

Zhu X., Song C., Guo Y., Shi F., Wang L.: $N_2O$ emissions and its controlling factors from the peatlands in the Sanjiang Plain, China Environmental Sciences, 33, 2228–2234, 2013.

Zona D., Oechel W.C., Kochendorfer J., Paw U K.T., Salyuk A.N., Olivas P.C., Oberbauer S.F.,

Lipson D.A.: Methane fluxes during the initiation of a large-scale water table manipulation experiment in the Alaskan Arctic tundra, Global Biogeochem Cycles, 23, doi

10.1029/2009gb003487, 2009.

---

## Author Response (AR2)

May 28, 2021

Dear Editor and Reviewer:

I very much appreciate your efforts and time in reviewing our manuscript.
According to your precious advice and suggestions, we have revised this manuscript thoroughly.
Response to each question from editors and reviewers were listed below.
Thank you very much for your precious time and tremendous efforts in reviewing and supporting this manuscript.

Best Regards,

Xinyu Liu
Inner Mongolia Key Lab of River and lake ecology & Ministry of Education Key Laboratory of Ecology and Resource Use of the Mongolian Plateau
School of Ecology and Environment
Inner Mongolia University
Room 106, Biology Building
No. 235, West University Road, Saihan District, Hohhot
Inner Mongolia 010021, P. R. China
Mobile: +86-13245131615
E-mail: 21815009@mail.imu.edu.cn

Editor's comments:

a) "Degradation of downstream wetlands has resulted in the loss of the soil carbon pool by approximately 60% " ......   reduced?

Reply: Yes, wetland degradation leads to a reduction in soil carbon. We have replaced the word "the loss of" with "reducing" from the sentence, and the sentence has been changed to "Degradation of downstream wetlands has resulted in reducing the soil carbon pool by approximately 60%.....".

b) "...... shifting the $CH_4$ and $N_2O$ emissions from the source to the sink". Could you formulate this more precisely? "converting the wetland from a $CH_4$ and $N_2O$ sink to a source"? Is that what you mean?

Reply: Generally, a healthy wetland is a $CH_4$ and $N_2O$ source, but $CH_4$ and $N_2O$ act as sink after the wetland is degraded. We have revised this sentence as follows:

"converting the wetland from a $CH_4$ and $N_2O$ source to a sink".

c) "Although wetlands cover only 4–6% of the terrestrial land surface, they contain approximately 12–24% of global terrestrial soil organic carbon (SOC), thus acting as carbon sinks. Moreover, they release $CO_2$, $CH_4$, and $N_2O$ into the atmosphere and serve as carbon sources". But they cannot be sink and source at the same time. What is the net effect? I think that should also be mentioned here.

Reply: The net effect of wetland is a carbon sink, because the carbon accumulation by plant's photosynthesis is higher than the consumption (plant respiration, animal respiration, and microbial decomposition) in the wetland. We have added this sentence as follows:

"Although wetlands cover only 4–6% of the terrestrial land surface, they contain approximately 12–24% of global terrestrial soil organic carbon (SOC), thus acting as carbon sinks. Moreover, they release $CO_2$, $CH_4$, and $N_2O$ into the atmosphere and serve as carbon sources. In general, the carbon accumulation by plant's photosynthesis is higher than the consumption (plant respiration, animal respiration, and microbial decomposition) in the wetland, thus the net effect of the wetland is acted as a carbon sink".

d) "However, the process will be inhibited when the temperature was too high or too low". .......is?

Reply: We have replaced the word "was" with "is".

"However, the process will be inhibited when the temperature is too high or too low".

e) "This result clearly showed that $CO_2$ contributed more than $CH_4$ and $N_2O$ to global warming". Do you mean that there were more $CO_2$ emissions than $CH_4$ and $N_2O$ emissions? If so, then please reformulate.

Reply: In the previous sentence, we have shown the high accumulated $CO_2$ emissions. Thus this sentence, we want to show that $CO_2$ emissions have a significant impact on the greenhouse effect than $CH_4$ and $N_2O$ emissions. We have revised this sentence as follows:

"Therefore, the total annual cumulative $CO_2$ emissions are high. This result clearly showed that the significant impact of $CO_2$ emissions than $CH_4$ and $N_2O$ emissions on global warming".

f) "The hydrology and soil properties showed more evident differences among the transects because the downstream zone was dry all year due to the presence of the Xilinhot Dam (Fig. 1)". I am not sure, can something be "more" evident?

Reply: The result in our study indicates that a significant change in hydrology and soil properties between different transects. The word "more" is indeed misused. We have removed the word "more".

"The hydrology and soil properties showed evident differences among the transects because the downstream zone was dry all year due to the presence of the Xilinhot Dam (Fig. 1)".

Reviewer's comment:

a) I would recommend to add few sentences in the conclusion about the overall $CO_2$ balance considering photosynthesis. I am aware that this balance could not be measured. However, it should be mentioned because otherwise the data could be misinterpreted (A wetland is a $CO_2$ sink in the overall $CO_2$ balance).

Reply: We have added a sentence in the conclusion about the overall $CO_2$ balance considering photosynthesis as follows:

"The riparian wetlands were the potential hotspots of GHG emissions in the Inner Mongolian region. However, the degradation of wetlands transformed the area from a source to a sink for $CH_4$ and $N_2O$ emissions, and reduced $CO_2$ emissions, which severely affected the wetland carbon cycle processes. Our results show that the riparian wetlands have high $CO_2$ emissions, but wetlands are $CO_2$ sink in the overall $CO_2$ balance general due to the photosynthesis of plants. Overall, our study suggests that anthropogenic activities have significantly changed the hydrological characteristics of the studied area, and will accelerate carbon loss from the riparian wetlands and further influence the GHG emissions in the future".

Other revisions:

a) We have added Heyang Sun as co-author to the manuscript. Heyang Sun is contributed much in the revised version of our manuscript. After we received the referee's comments, he provided many solutions and deeper insights on many issues and hence establish the contribution of this paper.

"Xinyu Liu[1,2], Xixi Lu[1,3], Ruihong Yu[1,2], Heyang Sun[1], Hao Xue[1], Zhen Qi[1], Zhengxu Cao[1], Zhuangzhuang Zhang[1], Tingxi Liu[4]".

b) We have updated the authors affiliations.

"[1] Inner Mongolia Key Laboratory of River and Lake Ecology, School of Ecology and Environment, Inner Mongolia University, Hohhot 010021, China;

[2] Key Laboratory of Mongolian Plateau Ecology and Resource Utilization, Ministry of Education, Hohhot 010021, China;

[3] Department of Geography, National University of Singapore, 117570, Singapor;

[4] Inner Mongolia Water Resource Protection and Utilization Key Laboratory, Water Conservancy and Civil Engineering College, Inner Mongolia Agricultural University, Hohhot 010021, China"

c) We have updated the fund in the Acknowledgements.

"This study was funded by the National Key Research and Development Program of China (grant no. 2016YFC0500508), Major Science and Technology Projects of Inner Mongolia Autonomous Region (grant nos. 2020ZD0009 and

ZDZX2018054), National Natural Science Foundation of China (grant no. 51869014), Key Scientific and Technological Project of Inner Mongolia (grant no.2019GG019), and Open Project Program of the Ministry of Education Key Laboratory of Ecology and Resources Use of the Mongolian Plateau (grant no. KF2020006)."

---

## Author Response (AR3)

June 22, 2021

Dear Editor:

I very much appreciate your efforts and time in reviewing our manuscript.

According to your precious advice and suggestions, we have revised this manuscript thoroughly.

Response to each question from the editor was listed below.

Thank you very much for your precious time and tremendous efforts in reviewing and supporting this manuscript.

Best Regards,

Xinyu Liu

Inner Mongolia Key Lab of River and lake ecology & Ministry of Education Key Laboratory of Ecology and Resource Use of the Mongolian Plateau

School of Ecology and Environment

Inner Mongolia University

Room 106, Biology Building

No. 235, West University Road, Saihan District, Hohhot

Inner Mongolia 010021, P. R. China

Mobile: +86-13245131615

E-mail: 21815009@mail.imu.edu.cn

Editor's comment:

a) There are still a few language issues and I would recommend to give the manuscript to a native speaker or a professional language editing service for a final language check.

Reply: We have given the manuscript to a professional language editing service for a final language check. Certificate of editing as follow:

**Wiley Editing Services**

**ENGLISH EDITING CERTIFICATE**

This document certifies that the manuscript listed below was edited for proper English language, grammar, punctuation, spelling, and overall style by one or more of the highly qualified native English speaking editors at Wiley Editing Services

**Manuscript title**

Greenhouse gases emissions from riparian wetlands: An example from the Inner Mongolia grassland region in China

**Authors**

Xinyu Liu, Xixi Lu, Ruihong Yu, Heyang Sun, Hao Xue, Zhen Qi, Zhengxu Cao, Zhuangzhuang Zhang, Tingxi Liu

**Order No**
YVZLT_1_3

**Date Issued**
June 21, 2021

[Figure]

This document certifies that the manuscript listed above was edited for proper English language, grammar, punctuation, spelling, and overall style. Neither the research content nor the authors' intentions were altered in any way during the editing process. Documents receiving this certification should be English-ready for publication; however, the author has the ability to accept or reject our suggestions and changes. If you have any questions or concerns about this document or certification, please contact help-cn@wileyeditingservices.com.

Wiley Publishing Services is a service of Wiley Publishing. Wiley's Scientific, Technical, Medical, and Scholarly (STMS) business serves the world's research and scholarly communities, and is the largest publisher for professional and scholarly societies. Wiley is committed to providing high quality services for researchers. To find out more about Wiley Editing Services, visit http://wileyeditingservices.com. To learn more about our other author services provided by Wiley Publishing, visit https://authorservices.wiley.com/

**WILEY**

---

## Author Response (AR4)

August 2, 2021

Dear Editor:

I very much appreciate your efforts and time in reviewing our manuscript.

According to your precious advice and suggestions, we have revised this manuscript thoroughly.

Response to each question from the editor was listed below.

Thank you very much for your precious time and tremendous efforts in reviewing and supporting this manuscript.

Best Regards,

Xinyu Liu

Inner Mongolia Key Lab of River and lake ecology & Ministry of Education Key Laboratory of Ecology and Resource Use of the Mongolian Plateau

School of Ecology and Environment

Inner Mongolia University

Room 106, Biology Building

No. 235, West University Road, Saihan District, Hohhot

Inner Mongolia 010021, P. R. China

Mobile: +86-13245131615

E-mail: 21815009@mail.imu.edu.cn

Editor's comment:

a) Please insert an information regarding the copy-editing service to the acknowledgments. Moreover, can the supplement be deleted then? Please clarify.

Reply: We have inserted information regarding the copy-editing service into the acknowledgments. Moreover, the supplement can delete.

"This study was funded by the National Key Research and Development Program of China (grant no. 2016YFC0500508), Major Science and Technology Projects of Inner Mongolia Autonomous Region (grant nos. 2020ZD0009 and ZDZX2018054), National Natural Science Foundation of China (grant no. 51869014), Key Scientific and Technological Project of Inner Mongolia (grant no. 2019GG019), and Open Project Program of the Ministry of Education Key Laboratory of Ecology and Resources Use of the Mongolian Plateau (grant no. KF2020006). We thank Wiley Editing Services (http://wileyeditingservices.com) for its linguistic assistance during the preparation of this manuscript".